



# Global climate response to idealized deforestation in CMIP6 models

Lena Boysen[1], Victor Brovkin[1,2], Julia Pongratz[1,3], David Lawrence[4], Peter Lawrence[4],

Nicolas Vuichard[5], Philippe Peylin[5], Spencer Liddicoat[6], Tomohiro Hajima[7], Yanwu Zhang[8], Matthias Rocher[9], Christine Delire[9], Roland Séférian[9], Vivek K. Arora[10], Lars Nieradzik[11], Peter Anthoni[12], Wim Thiery[13], Marysa Laguë[14], Deborah Lawrence[15], Min-Hui Lo[16]

[1]The land in the Earth System, Hamburg, Germany.

[2]Center for Earth System Research and Sustainability, Universität Hamburg, Germany.

[3]LMU, Department of Geography, Munich, Germany

[4]Climate and Global Dynamics Laboratory, National Center for Atmospheric Research, Boulder, CO, USA

[5]Laboratoire des Sciences du Climat et de l'Environnement, Gif-Sur-Yvette, France

[6]Met Office Hadley Centre, Exeter, UK

[7]Research Institute for Global Change, Japan Agency for Marine-Earth Science and Technology,Yokohama, Japan

[8] Beijing Climate Center, China Meteorological Administration, Beijing, China

[9] CNRS, Université de Toulouse, Météo-France, Toulouse, France

[10]Canadian Centre for Climate Modelling and Analysis, Environment and Climate Change Canada, Victoria, BC, Canada

[11]Institute for Physical Geography and Ecosystem Sciences, Lund University, Lund, Sweden

[12] Karlsruhe Institute of Technology, Institute of Meteorology and Climate Research/Atmospheric Environmental Research, Garmisch-Partenkirchen, Germany

[13]Vrije Universiteit Brussel, Department of Hydrology and Hydraulic Engineering, Brussels, Belgium

[14]University of California, Berkeley, Department of Earth and Planetary Science, Berkeley, CA, USA

[15] Environmental Sciences, University of Virginia, Charlottesville, VA, USA

[16]Min-Hui Lo, Department of Atmospheric Sciences, National Taiwan University, Taipei, Taiwan.

*Correspondence to*: Lena R. Boysen (lena.boysen@mpimet.mpg.de)



**Abstract**

Changes in forest cover have a strong effect on climate through the alteration of surface biogeophysical and biogeochemical properties that affect energy, water, and carbon exchange with the atmosphere. To quantify biogeophysical and biogeochemical effects of deforestation in a consistent setup, nine Earth System models carried out an idealized experiment in the framework of the Coupled Model Intercomparison Project, phase 6 (CMIP6). Starting from their pre-industrial state, models linearly replace 20 million $km^2$ of tree area in densely forested regions with grasslands over a period of 50 years followed by a stabilization period of 30 years. Most of the deforested area is in tropics, with a secondary peak in the boreal region. This study compares the effect of this large deforestation perturbation on energy and carbon fluxes across models. The effect on global annual near-surface temperature ranges from no significant change to a cooling by 0.55°C, with a multi-model mean of -0.22±0.21°C. Five models simulate a temperature increase over deforested land in the tropics and a cooling over deforested boreal land. In these models, the latitude at which the temperature response changes sign ranges from 11 to 43°N, with a multi-model mean of 23°N. A multi-ensemble analysis reveals that the near-surface temperature changes emerge within 50 years over the tropical regions propagating from the centre of deforestation to the edges, indicating the influence of non-local effects. The biogeochemical effect of deforestation are land carbon losses of 259±80 PgC. Based on transient climate response to cumulative emissions (TCRE) this would yield a warming by 0.46±0.22°C, suggesting a net warming effect of deforestation. While there is general agreement across models in their response to deforestation in terms of change in global temperatures and land carbon pools, the underlying changes in energy and carbon fluxes diverge substantially across models and geographical regions. Future analyses of the global deforestation experiments could further explore the effect on changes in seasonality of the climate response as well as large-scale circulation changes to advance our understanding and quantification of deforestation effects in the ESM frameworks.

## 1. Introduction

Forests cover about 32 million $km^2$, or about a quarter of the ice-free land surface (Hansen et al., 2010). There are about three trillion trees on the Earth, most of them in the tropical and subtropical regions (Crowther et al., 2015). On local to global scales, tree-dominated ecosystems strongly affect land-atmosphere fluxes of water, energy, momentum (biogeophysical effects) and greenhouse gases (biogeochemical effects). A dominant driver of climate change effects is deforestation, as forest replacement with crops and pastures has a strong influence on land surface albedo (reflectivity) and transpiration, and leads to carbon losses to the atmosphere. Historical deforestation has amounted to 22 $Mkm^2$ between year 800 and 2015 and future forest losses could almost be that high, too (Hurtt et al., 2020) to free land for food or bioenergy production or timber use. To understand the impact of deforestation on climate and the carbon cycle is of major importance. While the biogeochemical effects of deforestation, associated with release of carbon to the atmosphere, always lead to a warming at the global scale, biogeophysical effects, associated with changes in energy fluxes, differ in direction and magnitude between tropical and boreal regions (Pongratz et al., 2010). In the tropics, a reduction in evapotranspiration after deforestation generally leads to local warming (Claussen et al., 2001; Lejeune et al., 2015). Boreal deforestation generally cools the climate due to increased land surface albedo during the snow season (Bonan, 2008), especially in the spring, when the snow-masking effect of forests strongly affects the net radiation at the surface (Brovkin et al., 2006). Climate consequences of temperate deforestation are intermediate, with possible cooling in spring but warming in summer (Betts, 2000).

Biogeophysical effects of forest cover changes can be studied by using different model setups. As oceans cover most of the planet they dominate the response of the global temperature to any changes in boundary conditions. Experiments with interactive oceans



and sea ice (Brovkin et al., 2009; Davin and de Noblet-Ducoudré, 2010) as well as with slab oceans (Laguë et al., 2019) have

shown a global response of changes in climate in response to changes in forest cover. The sea ice – albedo feedback amplifies the response to a given external change, especially for boreal deforestation (Bala et al., 2007). Global effects of tropical deforestation are less certain, with effects of reduced water vapor generally leading to cooling of the atmospheric column (Ganopolski et al., 2001), while remote effects on atmospheric circulation are difficult to track (Lorenz et al., 2016). For example, teleconnections between tropical deforestation and precipitation over temperate North America could operate via the propagation of Rossby waves

(Medvigy et al., 2013). An experimental setup with atmosphere-only models in which sea surface temperatures are prescribed allows to increase the signal-to-noise ratio of models' response to deforestation. This setup assumes that the effect of large-scale circulation changes is small and can be ignored. Climatic effects of historical land use and land cover changes (LULCC) studied in this setup show substantial differences among global climate models due to differences in land surface schemes and their implementation of changes in land cover to represent deforestation (Boisier et al., 2012; de Noblet-Ducoudré et al., 2012; Pitman

et al., 2009).

Ideally, biogeophysical effects of deforestation are studied using a set of transient coupled simulations by comparing experiments with- and without deforestation (Brovkin et al., 2013; Lawrence et al., 2012). These studies require dedicated model experiments that are computationally costly. A less expensive approach is based on the idea of analysing differences in response of neighbouring

pairs of model grid cells that are deforested to different extents in the same numerical experiment (e.g. Kumar et al., 2013; Lejeune et al., 2018). This approach is well suited for post-processing results from existing experiments. It is also applied for analysis of remotely sensed data with pairs of grid cells that are affected differently by land cover changes (Alkama and Cescatti, 2016; Duveiller et al., 2018b; Li et al., 2015). Analysis of remote sensed data or any other analysis based on comparing grid cells with different vegetation cover under a similar climate, e.g. upscaled analysis of local fluxes (Bright et al., 2017) leads to different

interpretation of the effects of deforestation when compared to results from fully coupled model simulations. Typically, observation-based studies find a global warming in response to deforestation opposed to model simulations in which a global cooling dominates. Winckler et al. (2019a) showed that the reason for this discrepancy lies in the analyses of observation-based effects of deforestation which eliminate the non-local effects that propagate signals outside the location of deforestation by advection or changes in atmospheric circulation and constitute mostly a cooling for deforestation. Chen and Dirmeyer (2020)

confirmed for temperature extremes that accounting for atmospheric feedbacks could reconcile observations and model simulations.
Biogeochemical effects of deforestation are mainly quantified as losses of carbon storage in vegetation biomass and soils, but there can also be contributions from changes in the budgets of other greenhouse gases such as methane. As less above-ground carbon is stored in boreal ecosystems than in the tropics, boreal deforestation leads to less carbon losses per unit area than tropical

deforestation. Carbon losses depend also on what replaces the forest, cropland or grassland, and the post-deforestation land management practices such as fertilization and irrigation.

The Land Use Model Intercomparison Project (LUMIP; Lawrence et al., 2016) provides a unique opportunity to compare the sensitivity to deforestation for ESMs participating in phase 6 of the Coupled Model Intercomparison Project (CMIP6; Eyring et

al., 2016). This study focuses on the idealized global deforestation experiment (*deforest-globe*), an experiment within LUMIP framework, to investigate potential differences in the Earth System response in a setup combining boreal, temperate and tropical deforestation on a scale large enough to yield a significant signal-to-noise ratio. To limit the amount of simulations, the experimental protocol aims to combine both tropical and boreal deforestation in one scenario. As models have different forest





cover distributions in the pre-industrial control (*piControl*) simulation, the approach aims to remove the same amount of tree cover
area from the most forested grid cells. Branching off the *piControl* simulation, 20 million km$^2$ of forest are removed linearly over
a period of 50 years and replaced by grasslands. This is followed by a period of at least 30 years with no changes in forest cover
(see Fig. 2 in Lawrence et al. (2016) and Fig. S2). This setup is unique in that it induces a strong signal, i.e. aiming at robust
detection of modelled responses. Similar to the CMIP6 1% yr$^{-1}$ increase of $CO_2$ experiment, the model responses can be evaluated
over time in transient simulations. The main advantage, however, is the comparability of the model results due to a fairly simple
but harmonized deforestation specification compared to previous studies focusing on more realistic and diverse land cover changes
(Boysen et al., 2014; Brovkin et al., 2013).

Here, we analyse the response to this idealized deforestation scenario in nine ESMs participating in CMIP6. We first focus on the
biogeophysical effects, which manifest at local and non-local scales. This is underlined by in-depth analyses including the temporal
development of climate responses including Time of Emergence (ToE), a new metric, Fraction of Emergence (FoE), and land-
atmosphere coupling strength (Surface Energy Balance, SEB). Next, we analyse the changes in land carbon pools due to
deforestation and provide insights into different model formulations. These results provide insight into LULCC processes that
effect climate and their representation in the state-of-the-art models, but also have important implications for areas experiencing
rapid deforestation today.

## 2. Methods

### 2.1 Simulation set-up:

The *deforest-globe* experiment is described in detail in Lawrence et al. (2016) and summarised here briefly. In *deforest-globe*, land
use (land exploited by humans), land management (ways humans exploit the land), $CO_2$ and all other forcings are kept constant at
their pre-industrial levels. The selection of grid cells for deforestation is based on the fractional tree cover in a given model's
*piControl* simulation. Note, that we here use the terms tree and forest fraction interchangeably, although they are defined
distinctively in reality. Top 30% of grid cells with highest fractional tree cover are considered for deforestation. Within 50 years,
20 million square kilometres (Mkm$^2$) of forest area are removed in a linearly increasing manner, at a rate of 400,000 km$^2$ per year.
After deforestation all above-ground biomass is removed from the system (thus not interfering with atmospheric $CO_2$
concentrations), while below-ground biomass is transferred to litter and soil carbon pools. These areas are then replaced by
grassland. To assure permanence of this change, dynamic vegetation modules should be switched off over deforested areas in this
experiment. To allow the system to equilibrate, the simulation is run for at least 30 years following the end of deforestation, referred
to as the stabilization period.

In combination with the corresponding *piControl* simulation for each model, we can analyse the biogeophysical effects in this
experiment, as only changes in physical land surface properties can impact climate in this model formulation. While the effects of
deforestation on the various land carbon pools can be assessed during the deforestation and stabilization period, the carbon released
to the atmosphere is not "seen" by the atmosphere and therefore doesn't affect the climate and vegetation.

### 2.2 Models:

Nine ESMs carried out the deforest-globe experiment: MPI-ESM-1.2.0 (MPI, Mauritsen et al., 2019), IPSL-CM6A (IPSL, Boucher
et al., n.d.; Lurton et al., 2020), CESM2 (Danabasoglu et al., 2020), CNRM-ESM2-1 (CNRM, Delire et al., 2020; Séférian et al.,



2019), CanESM5 (CanESM, Swart et al., 2019), BCC-CSM2-MR (BCC, Li et al., 2019), MIROC-ES2L (MIROC, Hajima et al., 2020), UKESM1-0-LL (UKESM, Sellar et al., 2019) and EC-Earth3-Veg (EC-Earth, (Doescher, n.d.; Hazeleger et al., 2012). A detailed description of the model components and simulation specifications relevant to this are provided in the supplement S1 and
Table S1. All models simulated the dynamic interactions between the land, the atmosphere and the ocean dynamics while keeping all external forcings except for the deforestation constant. Data from the *deforest-globe* and the *piControl* simulations were downloaded from the Earth System Grid Federation (ESGF; https://esgf.nci.org.au). Results from MIROC and EC-Earth were not available on ESGF at the time of the analysis.

Due to their model structure, some ESMs had to diverge from the simulation protocol as described hereafter. MIROC does not simulate a specific forest fraction and instead implemented the replacement of primary to secondary natural vegetation, which allows for regrowth of forests. EC-Earth implemented the deforestation by introducing primary to secondary land use transitions on the forested natural land area and switched off the dynamic tree establishment in the newly generated secondary land areas. In UKESM deforestation is implemented in a way that "woody" vegetation comprising trees and shrubs is converted to agricultural
grassland. Dynamic vegetation processes continued to allow the trees and shrubs to compete for space in the remaining natural part of the grid cell, but only allowed C3 and C4 crop and pasture PFTs to compete within their prescribed areas of the agricultural region. In CanESM above ground biomass is not removed from the system but is instead transferred to product, litter and soil carbon pools; hence, we only analyse vegetation and not soil and total land carbon changes for CanESM. We further exclude BCC from the analysis of litter, soil and total land carbon pools as root biomass from trees was removed with deforestation and not
transferred to the litter carbon pools. In IPSL, deforestation was implemented by selecting the greatest forested areas opposed to the largest forest fractions, shifting the focus to the lower latitudes where grid cell sizes are larger.

While most models provided one realization of the experiment, IPSL and CESM2 conducted three ensemble members and MPI seven. Further, MPI and MIROC continued the simulation for 70 years and CanESM for 10 years beyond the required 30 years
after the end of deforestation.

**2.3 Methodology:**

All spatial plots presented in this study show the running mean centred over the last 30 years of the simulation for climate variables (year 50 to year 79), and over the last 10 years for carbon variables (year 70 to year 79), thereby representing conditions at the end of the required stabilization period. Accordingly, the first 30 years for climate and 10 years for carbon variables from the *piControl*
simulation after branching-off the deforest-globe simulation were used as a reference period (see Table S1 for the branching year). Only areas with statistically significant changes at the 5% significance level are shown based on a modified *Student's t*-Test accounting for auto-correlation (Lorenz et al., 2016; Zwiers and von Storch, 1995). Contours show the area of deforestation that exceeds 0.001% of the grid cell until the end of the deforestation period. The analyses are done globally including all land and ocean or limited to the areas of deforestation as shown by the contours in the spatial plots. Zonal means or sums are smoothed by
an approximated 10° running mean by including as many grid cells as are captured by 10° in latitude to avoid geospatial regridding of data.

The Surface Energy Balance (SEB) decomposition approach is used to infer the contribution of changes in energy fluxes to changes in the surface temperature ($T_{surf}$) (e.g. Luyssaert et al., 2014). Through the Stefan-Boltzmann law, changes in longwave radiation
emitted from the surface are directly linked to $T_{surf}$. We can therefore analyse by how changes in the net shortwave radiation ($\Delta$net



shortwave; incoming minus outgoing, with outgoing being dependent on changes of the surface albedo), the incoming longwave radiation (Δincoming longwave) and changes in the latent (Δlatent) and sensible (Δsensible) heat fluxes contribute to the changes in $T_{surf}$ (eq. 1). Epsilon ($\varepsilon$) the surface emissivity, is assumed to be 0.97 (Hirsch et al., 2018a) and sigma ($\sigma$) is the Stefan-Boltzmann constant with $\sigma$=5.67 $10^{-8}$ W $m^{-2}K^{-4}$. $T_{surf,piControl}$ is the surface temperature of the *piControl* simulation. We further assume that the

long-term mean ground heat flux is approximately zero (Winckler et al., 2017). This method has been widely used to analyse the biogeophysical effects of land use, land management and land cover changes on the surface fluxes (e.g. Hirsch et al., 2017, 2018a, 2018b; Thiery et al., 2017; Winckler et al., 2017). We also show $T_{surf}$ as simulated by the models ($T_{surf\_model}$); the difference between both ($T_{surf\_model}$ and $T_{surf}$) could thus hint to subsurface heat storage, non-negligible ground heat fluxes or changing emissivity (Broucke et al., 2015) as well as increased variability in $T_{surf\_model}$ that is not captured by $T_{surf}$.


$$\Delta T_{surf} = \frac{1}{4\varepsilon\sigma Tsurf,piControl^3} (\Delta net\ shortwave\ +\ \Delta incoming\ longwave\ -\ \Delta latent\ -\ \Delta sensible) \qquad (1)$$

While the SEB approach concentrates on the surface temperature ($T_{surf}$), we provide zonal means and spatial and temporal plots for changes in near surface air temperature at 2m (ΔTas). ΔTas is chosen in accordance with previous multi-model studies to allow

for intercomparison. However, Winckler et al. (2019a) point out that ΔTas and $\Delta T_{surf}$ might differ when looking at local responses to deforestation.

We use the concept of Time of Emergence (ToE) to assess in which year the signal of near-surface 2m temperature (ΔTas), total land carbon (ΔcLand) or gross primary productivity (ΔGPP) becomes robust, i.e. when its change is larger than the noise. This

concept has been widely applied by using a variety of methods for calculating both signal and noise (e.g. Abatzoglou et al., 2019; Hawkins and Sutton, 2012). Here we refer to the approach presented by Lombardozzi et al. (2014) and Schlunegger et al. (2019) to capture the ensemble dimension. The signal (defined as the mean of trends from year $t_0$ to $t_i$) and noise (defined as the standard deviation over the trends from year $t_0$ to $t_i$) are computed over the ensemble for every time step $t_i$. ToE is reached when the signal to noise ratio exceeds two (SNR ≥ 2). We also adapt the concept to Fraction of Emergence (FoE) which denotes the deforestation

fraction at which ToE of ΔTas, ΔcLand or ΔGPP is reached to provide a time-independent measure. Only three models provided multi-member ensembles with MPI providing eight and IPSL and CESM2 providing results from three ensemble members each.

The transient climate response to cumulative carbon emissions (TCRE, Gillett et al., 2013) identifies the amount of warming (ΔTas, relative to the pre-industrial state) per unit cumulative emissions at the time when atmospheric $CO_2$ concentrations double in the

1% $yr^{-1}$ $CO_2$ simulation. These ratios, expressed as °C $EgC^{-1}$ (1 exagram of carbon = $10^8$ gC), have been identified for a range of CMIP6 models by Arora et al. (2019): 1.6 (MPI), 2.24 (IPSL), 2.08 (CESM2), 2.21 (CanESM), 1.64 (CNRM), 1.3 (BCC), 1.32 (MIR) and 2.38 (UKESM) (no data available for EC-Earth). TCRE has been shown to give a good first estimate of ΔTas to ΔcLand changes in previous studies, (e.g. Arora et al., 2019; Boysen et al., 2014; Brovkin et al., 2013).





## 3. Results & Discussion

### 215 3.1 Deforestation patterns:

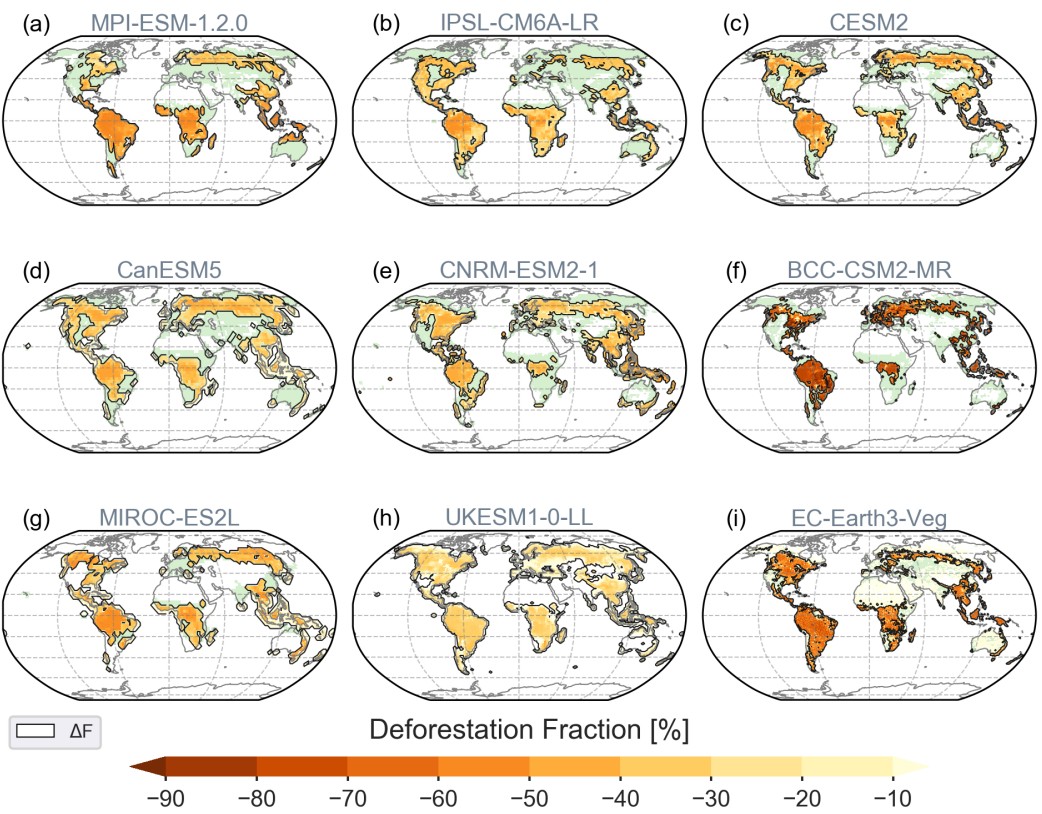

**Figure 1** Deforestation fractions ΔF in % of the grid cell area after the forced forest clearing is finished, shown in orange; green colors display the remaining forest extent. A map of the initial forest fractions can be found in the supplement (Fig. S1 and S2). Contours of the deforestation areas (ΔF) with deforested grid cell fraction exceeding 0.001% are used in all maps of the analysis.

We retrieved deforestation patterns (Fig. 1) by taking the difference of forest fraction between the *deforest-glob* at year 80 and the *piControl* simulation (ΔF), with a few exceptions noted here. Since dynamic vegetation was still switched on outside the deforested areas in UKESM, we considered tree cover changes only until year 50 to exclude forest changes afterwards that origin from outside the study area. For EC-Earth and CNRM a separate file was provided to identify deforestation fractions based on prescribed land cover changes. For BCC and CESM2, we subtracted the first timestep from the *deforest-glob* simulation because the required

variable treeFrac was missing in the *piControl* simulation. MIROC does not simulate specific forest cover and therefore provided a separate deforestation map based on prescribed land cover changes replacing primary with secondary vegetation; regrowth of forest could not be suppressed.

Deforestation of the top 30% grid cells with regard to their forested fraction in the *piControl* simulation of 1850 (see Table 1)

leads, as expected, to the largest forest removal in the tropical and boreal zone across all models (Fig. 1 and Fig. S2). Regional differences in the spatial pattern of deforestation across models can mainly be attributed to differences in the initial forest cover



(36 to 66 Mkm$^2$, Table 1) which is, for instance, almost twice as large in CNRM compared to EC-Earth and BCC. UKESM, CanESM and CESM2 generally remove more than twice as much forest in boreal regions compared to MPI and BCC in North America or IPSL and MIROC in Eurasia. MPI simulates less initial forest cover in temperate regions, in contrast, especially to CanESM, BCC, EC-Earth and CNRM. In EC-Earth, MPI and IPSL, tropical deforestation dominates the global patterns. The spread in initial forest cover highlights the difficulty in implementing any given land use and land cover change scenario (Di Vittorio et al., 2014). Overall, all models successfully perform 20 Mkm$^2$ (range 19.6 - 21.6 Mkm$^2$) of deforestation after 50 years.

The reconstructed potential forest cover is estimated to be 48.68 Mkm$^2$ in 800CE or 45.65 Mkm$^2$ in 1700 (Pongratz et al., 2008). The multi-model mean initial forest cover area of 48.22 Mkm$^2$ in Table 1 compares reasonably well with this estimate. The area deforested in the *deforest-globe* scenario is comparable to the historical deforestation area of 22 Mkm$^2$ between year 800 and 2015, the increase of grazing land from 1850 to 2015 by 20.5 Mkm$^2$, and the projected forest loss of 20.3 Mkm$^2$ between 2015 and 2100 in the land-use scenario of SSP5 RCP8.5 (Hurtt et al., 2020). However, in the *deforest-globe* experiment deforestation occurs over a much shorter period of time and the geographical locations of deforestation differ.





**Table 1:** Changes in the mean state by the end of the *deforest-globe* simulation globally (both land and oceans) and over areas of deforestation (ΔF) alone. Values in parenthesis denote statistically non-significant values. Values in square brackets for MPI,
MIROC and CanESM denote values at the end of simulation (MPI and MIROC at year 150 and CanESM at year 90). Zero-lat denotes the latitude where the change in temperature in response to deforestation turns from temperate/boreal cooling to tropical warming applying an approximated 10° running mean. TCRE values [°C EgC⁻¹] from Arora et al. (2019) applied to global ΔcLand. cLand refers to the sum of cSoil, cVeg and cLitter. – denotes non-significant changes.

\*based on t0 of the *deforest-glob* simulation. \*\**separate file.* \*\*\* including statistically significant as well as non-significant
values. ''only accounting for ΔcVeg in the absence of ΔcLand.

| Model | Initial forest cover [Mkm²] | ΔTas global [°C] | ΔTas over ΔF [°C] | Zero lat of ΔTas | ΔPr [mm/yr] | ΔPr over ΔF [mm yr⁻¹] | ΔcLand [GtC] | ΔcLand over ΔF [GtC] | ΔTas (°C) to ΔcLand using TCRE |
|---|---|---|---|---|---|---|---|---|---|
| **MPI** | 48.15 | (-0.04) | (0.05) | 17.7°N | -7 | -108 | -315 (-345) | -317 [-350)] | 0.50 [0.55] |
| **IPSL** | 56.25 | (0.02) | (0.00) | 11.4°N | -5 | -40 | -187 | -184 | 0.42 |
| **CESM2** | 46.98* | -0.20 | -0.33 | 26.9°N | -5 | -11 | -342 | -342 | 0.71 |
| **CanESM** | 56.48 | -0.55 [-0.51] | -0.92 [-0.88] | 4.2°N | -12 | -53 | -169'' | -165'' | 0.37'' |
| **CNRM** | 66.39** | -0.29 | -0.70 | NA | -4 | -10 | -227 | -233 | 0.37 |
| **BCC** | 35.96* | -0.11 | (0.04) | 34.2°N | -5 | -24 | -185'' | -192'' | 0.24'' |
| **MIROC** | 40.86** | (-0.01) | (-0.01) | NA | (0) | (-6) | -128 [-113] | -137 [-143] | 0.17 [0.15] |
| **UKESM** | 45.53 | -0.51 | -1.03 | NA | -16 | -67 | -365 | -359 | 0.87 |
| **EC-Earth** | 37.42** | -0.33 | -0.70 | NA | -3 | -17 | -247 | -246 | NA |
| **Model mean\*\*\*** | 48.22 | -0.22 | -0.40 | 22.6°N | -6 | -37 | -259 | -260 | 0.46 |
| **Standard deviation\*\*\*** | 9.38 | 0.20 | 0.42 | 8.7 | 5 | 33 | 80 | 77 | 0.22 |

### 3.2 Biogeophysical effects:

The analysis of biogeophysical effects of deforestation is split into sections on global and regional changes in the mean state by
the end of the simulation period and the temporal evolution of the primary energy quantities.





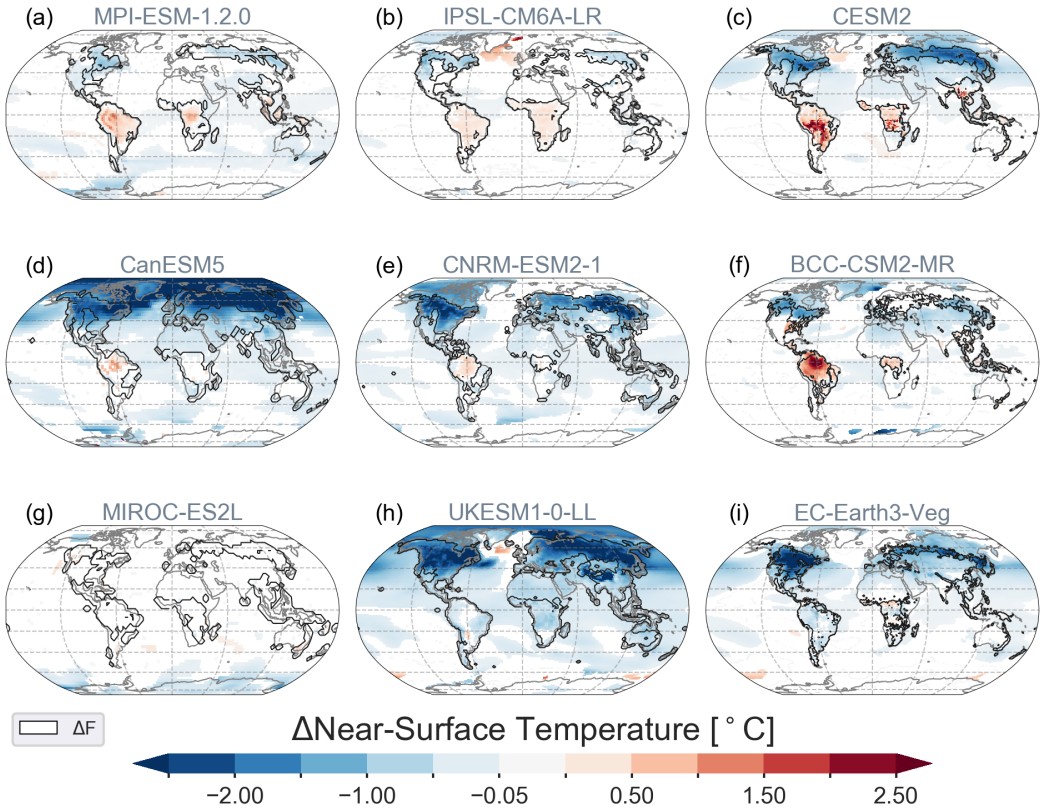

**Figure 2** Spatial patterns of near-surface air temperature responses. Only statistically significant changes at the 5% significance level are shown (modified *t*-test, Zwiers and von Storch, 1995). Contours depict the areas of deforestation (Fig. 1).




**Figure 3:** Zonally averaged surface energy balance (SEB) components after 80 years, expressed as contribution of changes in available energy (incoming and reflected shortwave and incoming longwave) and turbulent heat fluxes (latent and sensible) to changes in the surface temperature ($T_{surf}$). $T_{surf}$ is derived from the SEB decomposition method; $T_{surf\_model}$ as simulated by each model; The difference of $T_{surf\_model}$ and $T_{surf}$ represents the residual flux accounting for the ground heat flux and subsurface heat

storage. For comparison, the simulated near-surface temperature ($\Delta Tas$) is shown as well. An approximated running mean over 10





degrees latitude was applied to smooth lines. Only changes over deforested grid cells are shown; changes over all areas including all land and oceans can be found in Fig. S3.

### 3.2.1 Changes in mean near surface temperature

Six of the nine models simulate a statistically significant decrease in global near-surface air temperature in response to large-scale

deforestation. Results of statistically significant changes (Table 1) range from -0.11 to -0.55 °C (multi-model mean -0.33°C) globally as simulated by BCC, UKESM, CanESM, CESM2, CNRM and EC-Earth, while MPI, IPSL and MIROC show no significant changes on the global scale. Over areas of deforestation (ΔTas over ΔF) the cooling is stronger (-0.33 to -1.03 °C, multi-model mean -0.40±0.42°C). Globally averaged, BCC simulates the weakest response as a consequence of balancing regional patterns (Fig 2). Globally and regionally, MIROC shows almost no response of ΔTas to the deforestation forcing since the

regrowing, secondary vegetation is very similar to the original land cover.

The global net decrease of air temperature is dominated by the changes over the oceans and in the Arctic (Fig 2). Using the Surface Energy Balance (SEB) decomposition approach we can analyse the contribution of varying energy fluxes to the change in surface temperature ($\Delta T_{surf}$, Fig. 3). $\Delta T_{surf}$ is directly related to the balance of surface energy fluxes. However, it might deviate from ΔTas

at two meters height (see also Winckler et al. (2019a)), and is therefore also shown in Fig. 3 (dashed black lines). Besides Fig. 3 we provide a component-wise SEB decomposition across models in Fig. S3 and show the contributing fluxes in Fig. S4 (including cloud cover and full and clear sky longwave radiation).

In the mid to high northern latitudes all models simulate an increase in albedo in response to deforestation which induces a cooling

(Fig. 3, red lines). This increase in albedo mainly originates from the reduction in snow-masking effect of forests allowing for a denser and longer lasting snow cover towards summer over grasslands that replace forests. Some models even simulate non-local effects: In CESM2, BCC and EC-Earth this effect is carried beyond the geographical regions of deforestation and in CanESM and UKESM the geographical extent of the cooling is amplified due to a positive sea-ice-albedo feedback over the Arctic Ocean (see Fig. S5). Longwave radiation is reduced across all models northward of 35°N mainly as a result of reduced surface temperatures

leading to less atmospheric trapping and re-emission of longwave radiation (Zeppetello et al., 2019). This effect dominates over the impact of increasing cloud cover over these latitudes in UKESM, EC-Earth, BCC and CNRM which contributes with a longwave warming (see Fig S4 for zonal fluxes, Fig S6 for total cloud cover and Fig S7 for downward longwave radiation). UKESM, IPSL and CESM2 produce a 'warming blob' in the North Atlantic which in turn enhances sea surface evaporation and latent heat fluxes possibly due to the increased moisture demand of the atmosphere.


All models simulate reduction in available energy (due to reduced net shortwave and incoming longwave radiation) over areas of temperate and boreal ΔF (Fig. 3, red, brown and yellow lines) which dominate the reduction in temperature response , leading to cooling. The effects of increasing albedo are stronger than reduction in longwave radiation (Fig. S5). While net shortwave radiation reduces, the incoming shortwave radiation increases north of 40°N because the reduced evapotranspiration lowers the atmospheric

water vapor content and this increases the transmissivity of solar radiation through the atmosphere. In the MPI and IPSL models, reductions in cloud cover contribute to enhancement of transmissivity. With less net radiative energy entering the system, less energy is available for the generation of turbulent heat fluxes (latent plus sensible heat). At these higher latitudes all models except CNRM simulate decreased latent heat fluxes as not only forests are replaced by less evapotranspirative grassland, but also the atmospheric moisture demand due to the surface cooling and less moisture supply by precipitation reduce this flux. Similarly, most



models simulate reduced sensible heat fluxes as a consequence of reduced surface roughness and weaker vertical mixing. Only MPI and MIROC increase the sensible heat flux to balance the greater temperature gradient between the surface and atmosphere following the roughness reduction which is possible as net shortwave radiation is not as much reduced as in other models.

The global-scale deforestation-induced cooling is only offset over tropical forests. Here, most models (with the exception of EC-
Earth and UKESM) show a warming over $\Delta F$ (Fig 2), since the reduction in evapotranspiration and the decreases in latent heat fluxes dominates the decrease the albedo due to replacement of forests by grasslands (Fig. 3). However, the geographical patterns differ across models. All models simulate an increase in albedo in the tropics as brighter grasses replaced the darker forests. However, more incoming shortwave radiation due to reduced cloud cover more than compensates the reduction in incoming shortwave radiation associated with an increase in albedo in IPSL, CanESM, CNRM and BCC. UKESM is the only model that
simulates tropical cooling at the surface and at 2m height, with reduction in incoming shortwave radiation due to increasing albedo more than compensated reduction in latent heat and roughness decreases leading to overall cooling. This dominant effect of albedo changes as also observed in HadGEM2-ES which shares similar model components (Robertson, 2019).

In CESM2, in the equatorial tropics, evaporation increases (Fig S8). This unintuitive response may be due to the fact that $C_4$
grasses, which were parameterized for dry regions, are over-productive when they replace forests in the moist deep tropics. This cooling effect is balanced by reduced sensible heat fluxes and increased net shortwave radiation due to less cloud cover resulting in a net warming. Similarly, in EC-Earth evaporative cooling prevails from 30°N to 50°S since unmanaged grasses show strong increases in leaf area that in turn increases the transfer of soil moisture to the atmosphere. However, this cooling is overcompensated by the strongest decrease across models and latitudes in sensible heat fluxes as a consequence of a very low
surface roughness of grasses. BCC simulates the strongest temperature increases over the tropical region across all models leading to a net increase in temperature averaged across all areas of $\Delta F$. In the Amazon region this is mainly caused by an initial surface drying due to reduced evapotranspiration and increased sensible heat flux. This strengthens the circulation over the northern Amazon, supporting increased vertical convection of hot air that in turn causes horizontal advection of moist air from the tropical Atlantic (note that evaporation from the land decreases over the northern Amazon). This leads to increased cloud formation, which
increases incoming longwave radiation (with all-sky surface longwave radiation being larger than clear-sky surface longwave radiation, not shown).

Comparing $\Delta Tas$ and $\Delta T_{surf}$ reveals that there can be large differences among both variables. In CNRM, the surface warming of $\Delta T_{surf}$ which is dominated by reduced evapotranspiration is not seen in $\Delta Tas$ at 2m height. In EC-Earth the effect of reduced sensible
heat fluxes causes warming at the surface (Fig. 3i) which is not mixed upwards to the 2m level (Fig 2i) where a cooling is observed. MPI and CanESM show the smallest deviations in $\Delta Tas$ and $\Delta T_{surf}$.

The SEB approach applied here neglects the ground heat flux on longer averaging periods, subsurface heat storage or changing emissivity. However, inferring the difference between modelled and analytically determined $T_{surf}$
we see remaining negative differences in energy fluxes at higher latitudes in IPSL, CNRM and EC-Earth (difference of $T_{surf\_model}$ and $T_{surf}$, Fig. 3). This deviation from the simplifying assumption of our SEB approach assuming zero changes in the above-mentioned properties and fluxes needs further investigation which is beyond the scope of this manuscript.



In four out of nine models that simulate tropical warming and temperate/boreal cooling in response to deforestation the switch in
sign of ΔTas from warming to cooling ranges from 11.4°N in IPSL to 34.2°N in BCC (multi-model mean 22.6°N) if changes in
ΔTas over ΔF (Fig 3, dashed lines) are zonally averaged (Table 1). This change in sign of the temperature response due to the
biogeophysical effect is an important metric which indicates that the biogeophysical effects of re/afforestation would result in
cooling south of this latitude, in addition to a cooling effect due $CO_2$ removal from the atmosphere. Because the other 5 models
show the cooling effect of deforestation at all latitudes due to non-local effects, this estimate is highly uncertainty with a standard
deviation of at least 10° (Table 1).

Overall, the local response over deforested areas is a reduction of available energy (net shortwave plus downwelling longwave
radiation, Fig S4a) across all models at higher latitudes (by -5 to -20 W m$^{-2}$ and -0.5 W m$^{-2}$ in MIROC) due to snow, and brighter
grasses versus darker trees, albedo feedbacks. At lower latitudes, the models' response is more diverse mainly due to differences
in cloud formation (-5 to +10 W m$^{-2}$). As a result of less energy being available, turbulent heat fluxes reduce across all models and
latitudes (by -1 to -12 W m$^{-2}$) (Fig S4b). Most models simulate decreased latent heat fluxes over less evapotranspirative grasses.
Only two exceptions were found where grass parametrizations lead to higher latent heat fluxes (CESM2 and EC-Earth). Most
models simulate a reduction in sensible heat fluxes over temperate and boreal over grasslands which replaces forests as roughness
over grasses is lower which weakens vertical mixing. These findings are in line with those of Winckler et al. (2019b) who find a
dominating role of surface roughness for local effects of deforestation. MPI and MIROC simulate an increase in sensible heat as
net radiation reductions in these models are small and energy is partitioned preferentially towards the sensible heat flux. In the
tropical region, stronger sensible heat fluxes are seen everywhere in response to deforestation where $\Delta T_{surf}$ increases due to a
stronger temperature gradient between the surface and the atmosphere. In CanESM, EC-Earth and locally in CESM2 and BCC,
however, the effect of a strongly reduced roughness outweighs the impact of the temperature gradient.


Previous studies on the temperature effects of large-scale or historical deforestation have shown that the locally induced changes
in albedo after boreal deforestation are almost balanced by concurrent changing turbulent heat fluxes. However, the increased
boreal albedo can also induce a non-local cooling over land and oceans via advection of cooler and dryer air (Chen and Dirmeyer,
2020; Davin and de Noblet-Ducoudré, 2010; Winckler et al., 2019a). Like in other multi-model studies on the biogeophysical
effects of deforestation it is difficult to separate local and non-local effects without further separation experiments. However, we
also find a mean cooling across all models globally and locally over the areas of deforestation. Only MPI, IPSL and BCC simulate
weaker non-local cooling effects and thus, almost balancing global mean temperature effects of tropical warming and boreal
cooling.

Still a key question is how models simulate the impact of deforestation on the turbulent heat fluxes (de Noblet-Ducoudré et al.,
2012; Pitman et al., 2009), depending not only on the plant-physiological behaviour (e.g., stomatal conductance, growing seasons,
leaf area index) but also on parametrizations of surface roughness and the soil hydrology schemes.





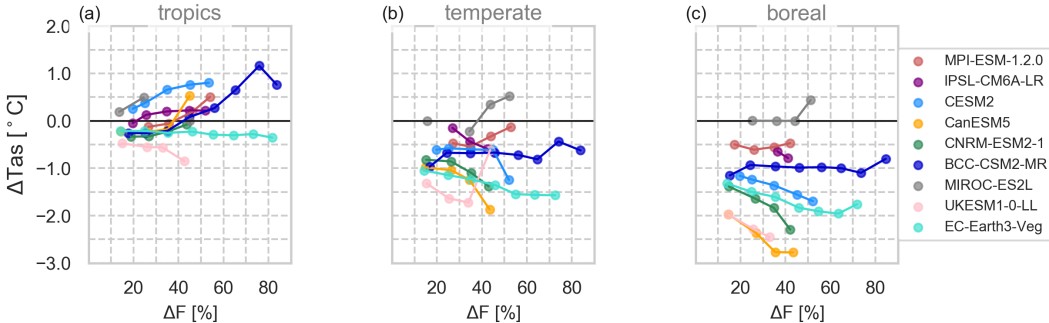

**Figure 4**: Relationship between temperature changes to the final deforestation fraction averaged over all pixels in the (a) tropical (23S to 23N), (b) temperate (50S to 23S and 23N to 50N) and (c) boreal (50N to 90N) region.

### 3.2.2 Forest sensitivity (FS) of ΔTas

The sensitivity of the models to the imposed deforestation signal by the end of the simulation period can be quantified in terms of the temperature change per unit fraction of grid cell deforestated (Fig. S9) or per unit area of deforestation (Fig. S10). We therefore
call it the 'forest sensitivity' (FS) to ΔF.

In the temperate and boreal regions, UKESM, CanESM and EC-Earth show temperature changes of more than -20°C frac$^{-1}$, CESM2, CNRM and BCC of -10°C frac$^{-1}$ and MPI, IPSL up to -4°C frac$^{-1}$. Per $10^3$km$^2$ of deforestation within one grid cell, UKESM and EC-Earth simulate temperature changes of more than -2.5 °C, CESM2, CNRM and BCC more than 1.3 °C, and CanESM, MPI and IPSL less than 0.6 °C per $10^3$km$^{-2}$. In the tropics, BCC and CESM2 show temperature increases of over 4°C
frac$^{-1}$, MPI and IPSL less than 2°C frac$^{-1}$ while EC-Earth and UKESM show decreases of up to -2° frac$^{-1}$. Per change in $10^3$km$^{-2}$ forest area, only BCC and EC-Earth show detectable changes of more than 0.5°C.

However, FS not only reflects local but also the superimposed non-local effects caused by feedback mechanisms. In the tropical region where non-local effects are smaller, we still see some differences in intensity. In particular, CESM2 and BCC and to a
smaller degree MPI reveal areas of stronger sensitivity to deforestation in the tropics than elsewhere (up to 8, 6 and 4 °C frac$^{-1}$, respectively). IPSL, CanESM and CNRM show smaller sensitivities and patterns of coupling also due to the superimposed non-local effects in the latter two models.

To draw more broad conclusions, FS is averaged for every 10% increase in ΔF per climate zone (Fig. 4). Over tropical regions
(23°S to 23°N, Fig. S11), MPI, CanESM, CNRM and BCC reveal increasing warming to increasing ΔF, which weakens and even stagnates in CESM2 and IPSL, respectively. On average these models show a tropical warming response of 0.27 °C frac$^{-1}$. UKESM and EC-Earth simulate increasing cooling with larger deforestation extent of -0.34 °C frac$^{-1}$. At higher latitudes (>50°N), five models show an increasing cooling with increasing ΔF (mean -1.31 °C frac$^{-1}$) which is increased by polar amplification. However, MPI, MIROC, BCC and EC-Earth show reverse tendencies at higher ΔF. Over temperate regions there is a more widespread
cooling (mean -0.78 °C frac$^{-1}$) due to mingling effects of different biomes, climate zones and generally smaller forest areas.

Previous studies have argued that the local temperature response to complete deforestation is stronger the smaller the initial forest cover was, and thus non-linear (Davin, 2016; Pitman and Lorenz, 2016; Winckler et al., 2017).



Only CESM2 and IPSL seem to produce the suggested non-linear, saturating behaviour over tropical regions where non-local

effects are smaller (see Fig 2) and a clear linear behavior cannot be found with any of the models.

However, drawing conclusions on the (non-) linearity is difficult. In our set up, ΔF reflects the top 30% of forested grid cells and thus links to the initial forest cover but without capturing the potential effects of completely cleared grid cells, smaller forest fractions, distinct ecozones or isolation from non-local feedback effects.

At a higher level of spatial precision including climate and ecozones, results like the ones presented here could be used to generate look-up tables for climate responses of each model to a given level of deforestation. These would provide computationally inexpensive tools to draw fast conclusions on the climate effects of deforestation in, for example, for future land-use scenarios. However, in some models the responses show a non-linear behavior not only to local coupling mechanisms but also due to climate feedbacks acting at the global scale. This superimposed, non-local signal should be isolated for models with strong Arctic

amplifications (here: CanESM, CNRM, UKESM, CESM2 and EC-Earth) to derive local climate responses. In addition, it would be preferable to use results from longer simulation periods once the models have equilibrated for such look-up tables.

### 3.2.3 Temporal analysis of ΔTas

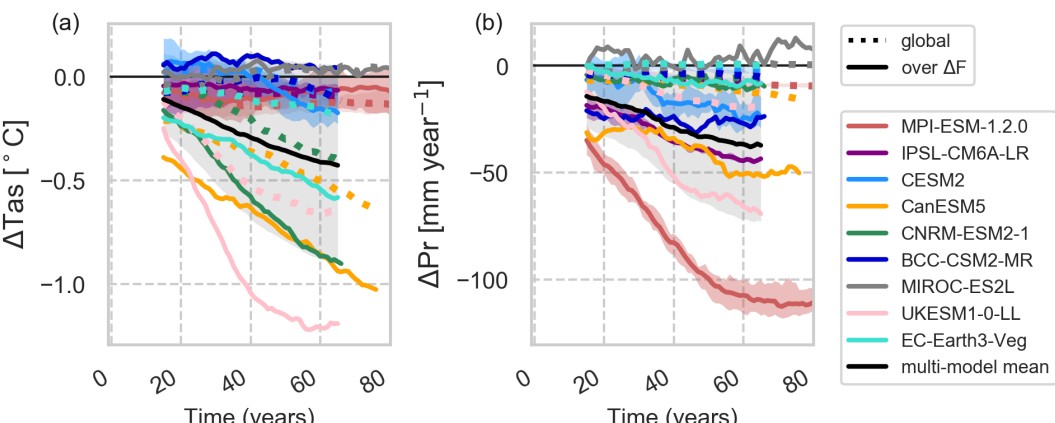

**Figure 5** Time series of temperature (a) and precipitation changes (b). Solid lines depict changes over areas ΔF while dotted lines depict global changes. A 30-year moving average is applied. The black line shows the multi-model mean with one standard deviation in shaded grey.

The results presented so far do not take into account whether the models have reached equilibrium by the end of the simulation

period. Globally, UKESM, CNRM, CanESM and EC-Earth simulate a linear response to the deforestation signal with CanESM, EC-Earth and CNRM showing a continuing downward trend after the end of deforestation, while UKESM stabilizes over ΔF and globally (Fig 5a). BCC simulates a more or less constant temperature increase over ΔF dominated by tropical warming, though the global signal is a slight cooling. MIROC drives hardly any change in ΔTas for any level of deforestation. MPI, IPSL and CESM2 show only small responses on the global scale due to balancing signals, but regionally, ΔTas scales with the intensity of

deforestation. Over South America, BCC, CESM2, MPI and IPSL simulate a linear increase while UKESM, CNRM and EC-Earth simulate decreases of Δtas with ΔF. In the boreal region, all models but MIROC simulate a linear decrease with ΔF over time, which clearly continues after 50 years in CanESM, EC-Earth and CNRM over North America and CanESM over Eurasia.



The temporal evolution of ΔTas reveals not only the sensitivity of models to large-scale deforestation but also the strength of non-
local high-latitude feedbacks. For most models it would have been beneficial to extend the simulation period to allow the climate
variables to reach equilibrium. The two models providing 150 years of data, MPI and MIROC, are less sensitive models without
strong feedbacks, and thus equilibrate quickly. For UKESM, recovering forests in the remaining parts of the deforested grid cells
shape the evolution of the signal.

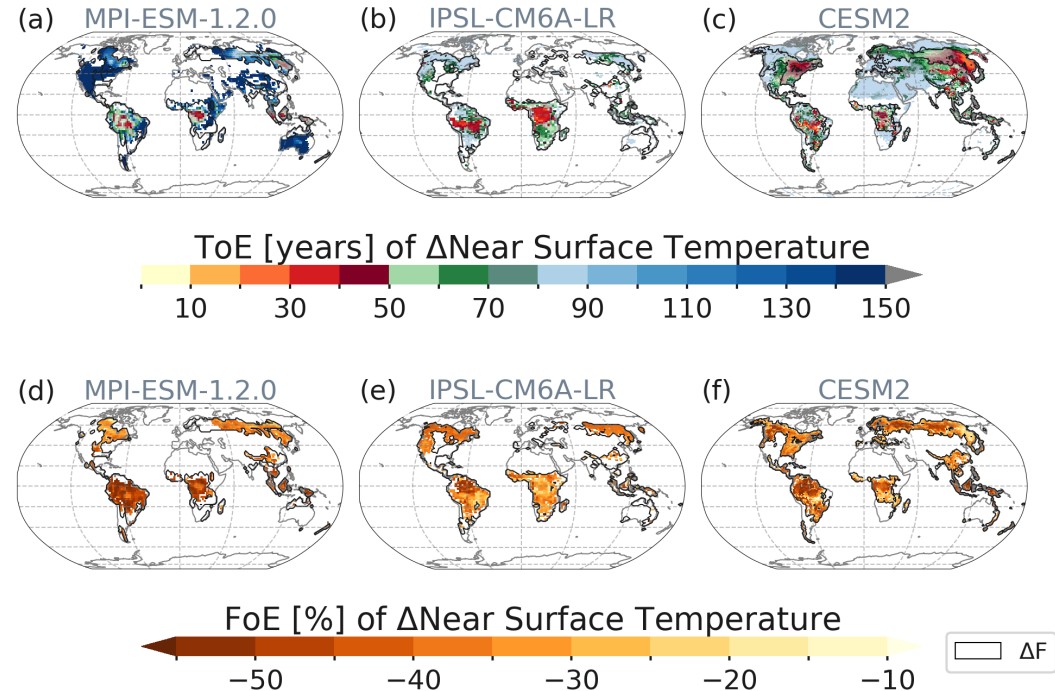


**Figure 6**: (a)-(c) Time of emergence (ToE) of ΔTas and (d)-(f) equivalent Fraction of emergence (FoE) of ΔTas. Only statistically
significant areas as found in Fig. 2 are shown; oceans are masked out.

For models that provided several ensemble members of the deforestation experiment (MPI, IPSL and CESM2) we calculated the
time of emergence (ToE). In the tropics, near-surface temperature changes emerge over the regions of strongest deforestation
before the end of the first 50 years of the simulation (Fig. 6). Interestingly, the signal propagates from the centre of deforestation
to the edges in the tropical zone. In the central tropics, the signal becomes robust (that is, exceeds the signal-to-noise ratio (SNR)
of 2) with up to 20% to 35% of deforestation still left (Fig S12). This hints to the advection of temperature changes towards the
centre of deforested area due to non-local effects. In boreal zones, CESM2, and to a lesser degree in MPI, demonstrate signals
propagating westwards starting from the boreal east coasts with about 30% and 10% of deforestation, respectively, still left. The
main attributor here are the westerly winds that carry the modified air by deforestation from the west to the east coast of the
continent where the signal is therefore strongest and emerges earlier. In CESM2, Arctic amplification further amplifies this process
(see section 3.2.1) leading also to responses outside ΔF. In MPI and IPSL, the advection of temperature changes from neighbouring
grid cells is limited to areas of ΔF. In the majority of areas, the signal takes more than 50 years to emerge despite the strong
imposed deforestation forcing (Fig. 6 green and blue colours).





The results of the ToE analysis have to be treated with caution since only a few ensemble members were available and thus, uncertainty remains high. However, following up on the earlier analysis (section 3.2.1), the observed patterns make sense from a

causal perspective. After 30 years, all three models demonstrate a propagation of signals from the center to the edges of ΔF and two show a westward propagation across the boreal zone. This emphasizes the importance of non-local biogeophysical effects during and after large-scale deforestation (Chen and Dirmeyer, 2020; Pitman and Lorenz, 2016; Winckler et al., 2019b). FoE is more universally applicable across models as the same amount of deforestation can happen at a different time in each model. Notably, while ToE patterns of ΔTas are very diverse across models, the FoE patterns are more alike, especially in the tropics.

These results lead to the conclusion that even after large-scale deforestation of one-third to half of the grid cell's forests, the signal only becomes robustly detectable after a few decades. The applied method based on ensemble trends gives an optimistic estimate of ToE compared to alternative approaches based on multi-ensemble or temporal means relative to the variability of a reference period (Fig. S13, note that SNR was lowered to 1).

Our results have important implications for ongoing land cover changes and climate policies. Between 2001 to 2018, 3.61 Mkm$^2$ of forests were cleared (Hansen et al., 2013) and http://earthenginepartners.appspot.com/science-2013-global-forest), thus the deforestation rate was about 20% of that applied in this study. Our results suggest that the detection of climate effects of this recent deforestation would possibly take decades. Likewise, climate response times to the reversal of deforestation as a mitigation measure are long compared to climate policy time scales.


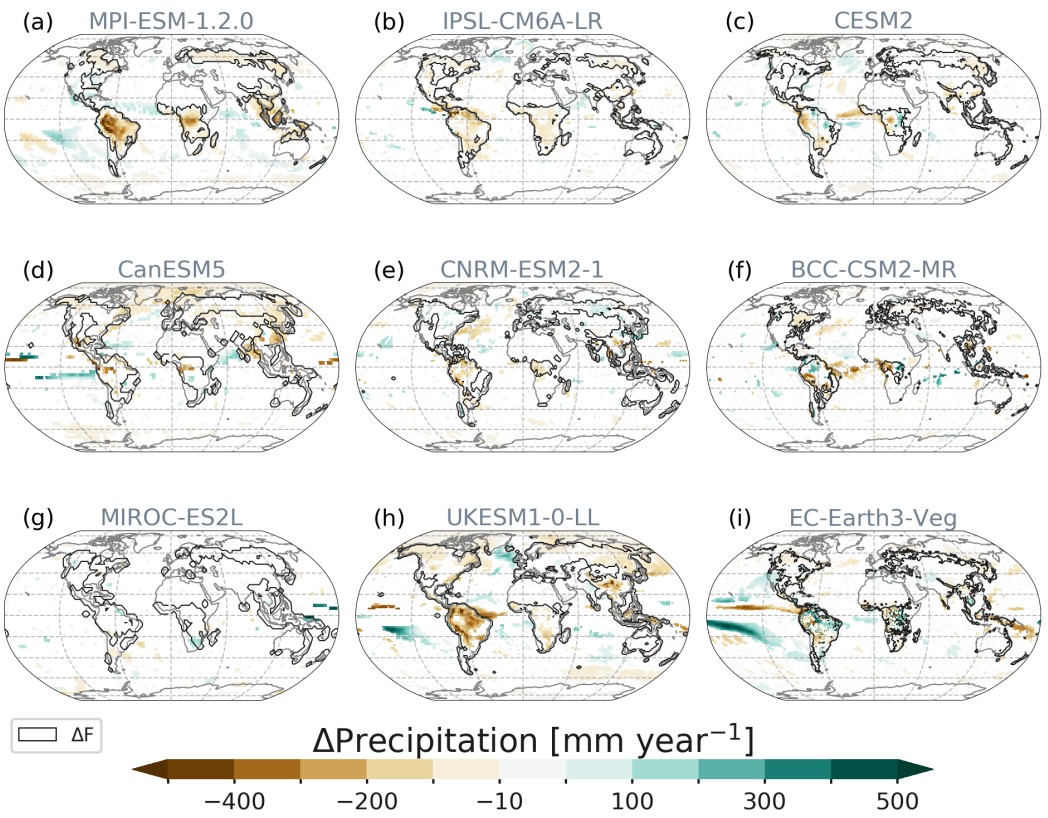



**Figure 7** Spatial patterns of precipitation responses. Only statistically significant changes shown. Contours depict the areas of deforestation (Fig. 1).

### 3.2.4 Changes in precipitation

The global net effect of precipitation changes over ΔF is negative across all models ranging from -10 to -108 mm year$^{-1}$ (mean -37±33 mm year$^{-1}$, Table 1). All models show shifts of atmospheric patterns over the oceans and distinct changes over ΔF (Fig 7). Again, MIROC is the least sensitive model with only minor increases over South Africa and Alaska.

The strongest global mean reduction of moisture transfer to the atmosphere via evapotranspiration (Fig. S8) and resulting
precipitation over ΔF is found in MPI, followed by UKESM, CanESM and IPSL. Generally, these decreases result from the replacement of forest by less evapotranspirative grasses. Precipitation increases occur mostly outside deforested regions but also on small scales over areas of ΔF in CESM2, CNRM, BCC and EC-Earth for different reasons: In CESM2, C4 grasses replace forest in tropical regions which are parametrized to be productive under unfavourable climate conditions and hence, are overly productive in tropical zones leading to transpiration increases. EC-Earth simulates increases in the tropics following increased
evapotranspiration (ET) due to a strong increase in leaf area. CNRM is the only model simulating precipitation increases over ΔF at northern latitudes providing moisture for enhanced evapotranspiration during snow-free months. In BCC, local vertical convection in the Amazon region causes horizontal advection of moist air from the Atlantic and west Amazon which locally increases precipitation there (see section 3.2.1).

Over tropical ΔF, most models show a linear relationship with relative temperature increases correlating with relative precipitation decreases and vice versa (Fig S14). UKESM shows a linear decrease of ΔPr with a decrease of ΔTas. Due to the above-mentioned model specifications, CESM2 and BCC show a very weak relationship between ΔTas and ΔPr over tropical ΔF with also positive ΔPr paired with positive ΔTas. Over boreal ΔF, most models simulate decreases in ΔPr correlated with decreases in ΔTas, while in CNRM ΔPr increases despite the cooling air.

Over time, MPI, UKESM and IPSL simulate linear responses to the deforestation signal with only MPI showing global stabilization after ending forest removal (Fig 5b). Other models exhibit longer time periods (CanESM and MIROC) of continuing positive or negative changes depending on the region. For example, over North America (not shown), CanESM simulates a downward and MIROC an upward trend in precipitation.

Global deforestation affects precipitation by altering circulation patterns and by changing the moisture inputs from the surface to the atmosphere. The SEB analysis (section 3.2.1) demonstrated how new plant types govern the land-atmosphere interaction via turbulent heat fluxes. In most cases we could infer a causal link between changes in turbulent heat fluxes, longwave radiation linked to cloud cover and precipitation which is in line with previous studies (e.g. Akkermans et al., 2014; Lejeune et al., 2015;
Spracklen et al., 2012). While most models simulate moisture decreases as less productive and evapotranspirative grasses replace trees, some models simulate local increases due to advected moisture (e.g., BCC) or favourable parametrizations of grasses (e.g., CESM2 and EC-Earth).



### 3.3 Biogeochemical changes

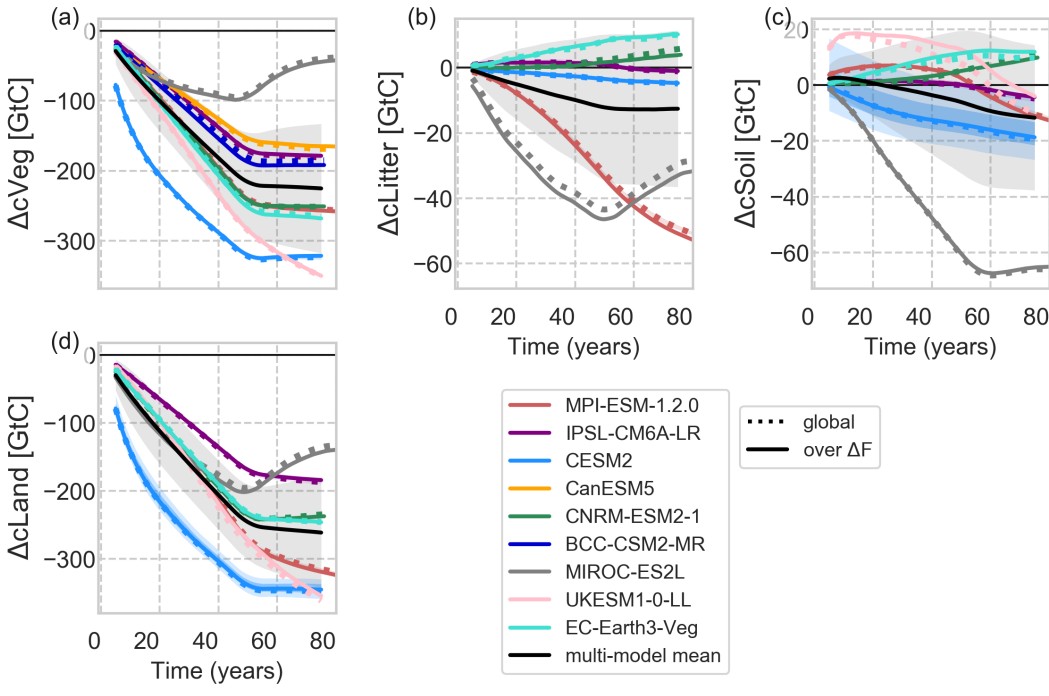


**Figure 8** Changes in carbon cycle pools over time smoothed by a 10-year moving average. Note that CanESM and BCC only ΔcVeg could be analysed. ΔcLand refers to the sum of ΔcSoil, ΔcVeg and ΔcLitter. The black line shows the multi-model mean with one standard deviation in shaded grey.



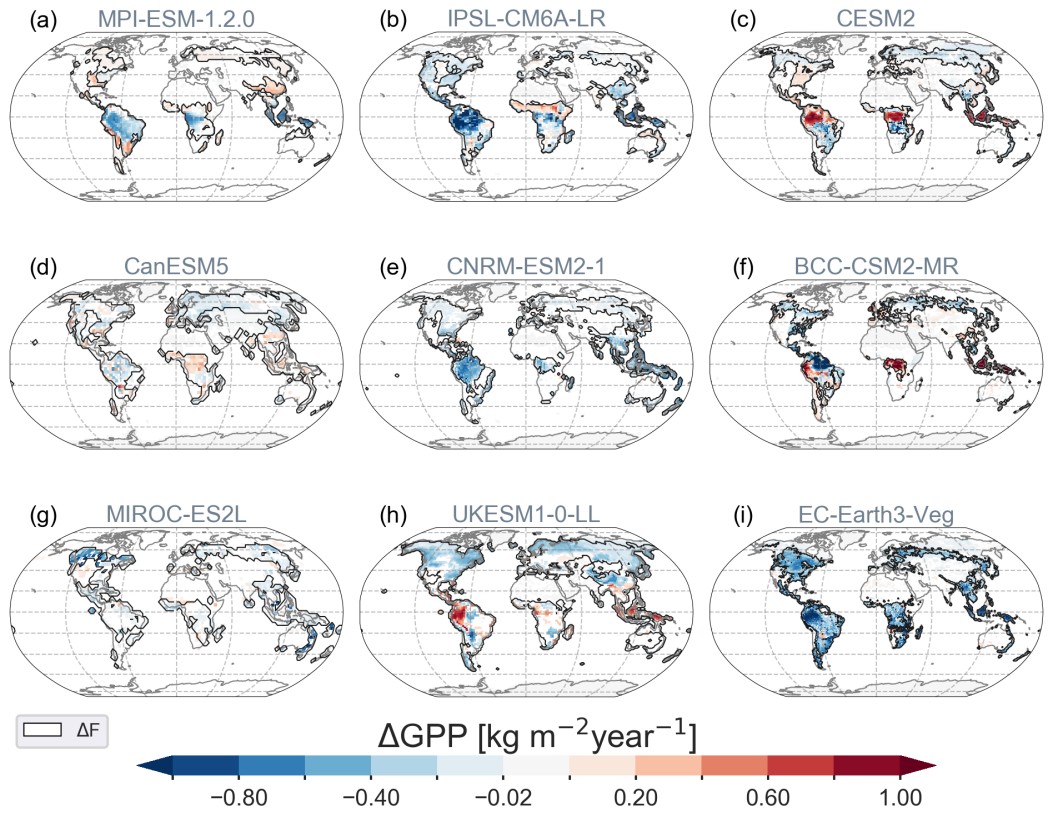


**Figure 9** Spatial patterns of ΔGPP responses. Only statistically significant changes at the 5% significance level are shown. Contours depict the areas of deforestation (Fig. 1).





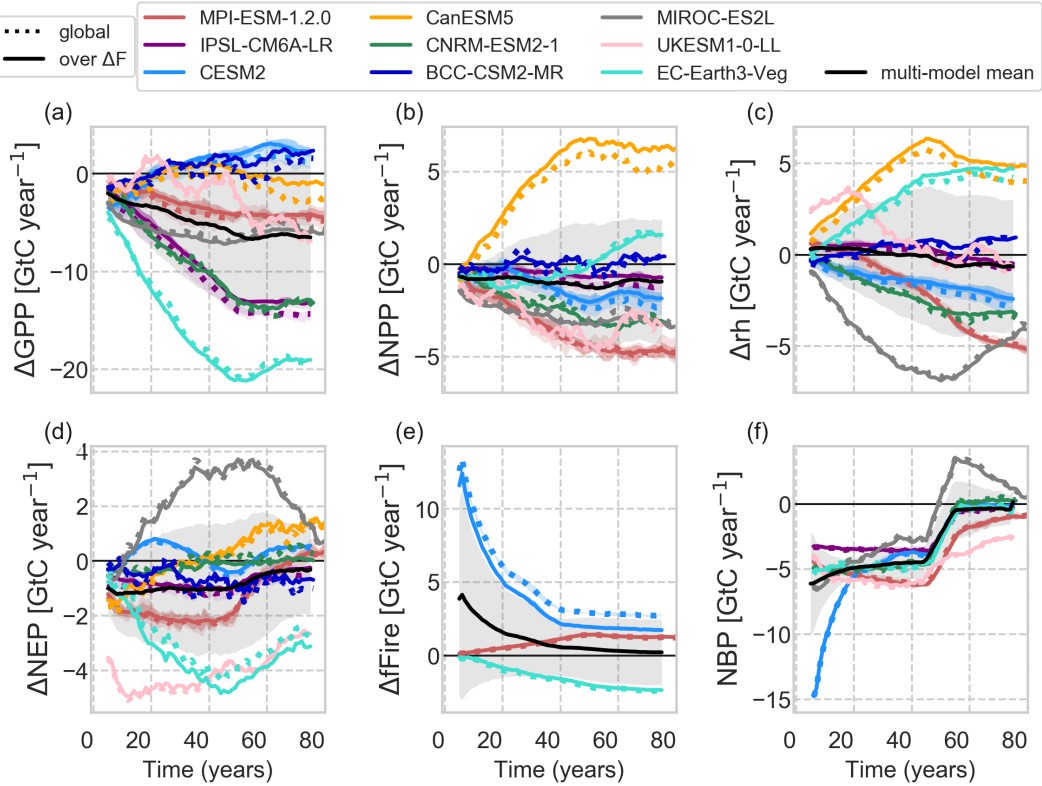

**Figure 10** Changes in carbon fluxes over time smoothed by a 10-year moving average. GPP and NPP are the gross and net primary
productivity, respectively; rh is the heterotrophic respiration, NEP is the net ecosystem productivity, fFire denotes the emissions
by fire and NBP the net biome productivity. NBP is based on year-to-year variations in ΔcLand and thus, not provided for CanESM
and BCC. The black line shows the multi-model mean with one standard deviation in shaded grey.

### 3.3.1 Changes in mean and temporal development of carbon pools and fluxes

Land carbon (cLand, the sum of vegetation, soil and litter carbon) losses range from -169 to -338 GtC over after 50 years of
deforestation are complete (Fig 8). Note, that CanESM and BCC were excluded in this calculation due to a major divergence from
the protocol. By the end of the experimental period the spread across models increases to -144 to -350 GtC with UKESM and MPI
simulating continuing declines and MIROC simulating increases. The multi-model mean decreases from -191 GtC after 50 years
(mean over year 45 to 54) to -203 GtC after 75 years (mean over year 70 to 79, Table 1) after the start of the simulation. The spatial
patterns of ΔcLand are displayed in Fig. S15. The spread across models is the result of several factors that make the model behave
differently.

For all models but MIROC, it is mainly the change in vegetation carbon (ΔcVeg) dynamics that dominate changes in cLand
followed by changes in litter carbon (ΔcLitter) and soil carbon (ΔcSoil).

In UKESM dynamic vegetation adjustments in the remaining natural parts of the deforested grid cells drive the continuing decline
and weak recovery of ΔcVeg as a consequence of strong cooling followed by stabilizing climate, respectively (Fig 8a and b).





Below ground carbon is transferred to the fast soil carbon pool (ΔcSoilFast, residence time of a year) which reduces with ongoing deforestation. Fast soil carbon decays and thereafter accumulates in the medium soil carbon pool (ΔcSoilMedium, residence time of several decades). The development of cLitter and heterotrophic respiration (rh) reductions and the subsequent development of all soil carbon pools correlate with the progression of deforestation and vegetation recovery afterwards. Note, that for UKESM below ground carbon from coarse roots is removed from the system and not transferred to the soil carbon.

In MIROC, because the vegetation type was fixed as woody types in the deforestation, forest recovery started soon after the deforestation process. As a result, the magnitude of ΔGPP is moderate among the models and ΔcVeg recovers as secondary woody vegetation, leading to the positive large NEP.

CESM2 shows a steep decline in cLand which is mainly caused by the initial vegetation loss and enhanced fire activity (carbon emissions by fire, ΔfFire) during deforestation of tropical forests due to degradation fires from deforestation (Li and Lawrence, 2017). The sudden initiation of deforestation fires in the Tropics contributes to emissions of 12 GtC yr$^{-1}$ at the beginning of the deforestation period and levelling off at around 1.7 GtC yr$^{-1}$ after 50 years (Fig 10 e) with highest values in the tropics (Fig S16). These initial deforestation fires cause GPP to drop for the first two decades before the overly productive C4 grasses in CLM5 (Lawrence et al., 2019), especially in the deep tropics (Fig 9) start to compensate for the carbon losses. ΔcLand saturates around -342 GtC with a minor negative drift after deforestation stops. The path is slightly nonlinear as the tropical grasses lead to recovery in ΔcLand after deforestation stops. In combination with a constant decline of heterotrophic respiration, net ecosystem productivity (ΔNEP) turns slightly positive by the end of the simulation period.

The slight continuing downward trend of ΔcLand in MPI is dominated by changes in the tropics (not shown). As grasses replace trees, ΔNPP is reduced strongly and consequently litter pools are reduced as well (in fact, MPI has the strongest NPP reduction across models with up to -2.2 GtC yr$^{-1}$ found in the tropics). Grass litter flux is not only smaller in amount, but also of changed quality leading to faster decomposition. Like in UKESM, the gain of fast soil carbon pool due to root decomposition during the first 50 years has only a minor effect in the long term (Fig. S17). Fire activity is fostered globally as grasses are more fire-prone than trees. In MPI, fire activity is enhanced because of a warmer tropical and globally drier climate (Fig. 10e), slowly diminishing land carbon pools at a similar rate as in CESM2 (~1.3 GtC yr$^{-1}$). Regional precipitation reductions cause heterotrophic respiration to decrease even more strongly than ΔNPP resulting in positive ΔNEP by the end of the simulation period.

EC-Earth and CNRM seem to behave similarly in terms of ΔcVeg, ΔcLitter and ΔcSoil at the global scale. However, regionally the models show fundamentally different responses. EC-Earth simulates higher deforestation rates in the tropics with subsequent higher cVeg loss than in CNRM and vice versa for higher latitudes. Interestingly, ΔcLitter and ΔcSoil increase in EC-Earth globally and CNRM in higher latitudes. In CNRM, grasses produce more belowground litter fall than trees (due to a higher root-to-shoot ratio) which accumulate, accompanied by lower overall Δrh fluxes, in all soil carbon pools.
In EC-Earth, increases in cLitter and cSoil are partly caused by the deforestation itself since portions of root and, against protocol, leaves and wood biomass and are left on site for decay. In addition, reductions in autotrophic respiration of grasses more than compensate GPP losses due to deforestation leading to a positive ΔNPP and thus, more litter. This litter is further contained as fire emissions in this model are reduced compared to the previous forest landscape (~-2 GtC yr$^{-1}$ globally). Even the substantial heterotrophic respiration increases due to local moisture input combined with mild cooling can therefore not deplete cSoil.





IPSL shows the smallest response in land carbon which is dominated by cVeg changes and hardly by any changes in soil or litter carbon pools. The exception is in central Africa where the higher NPP of grasses increases the litter flux affecting mainly the long-term soil carbon pool.


In BCC, tropical changes dominate the global average with the highest observed ΔGPP across models which is, however, diminished by a similarly high soil respiration flux resulting in a negative global ΔNEP throughout the simulation period. BCC is the only model that simulates cVeg increases outside ΔF in the temperate regions where cooling and precipitation increases overlap leading to a higher ΔGPP.


From CanESM, we only investigate ΔcVeg and carbon fluxes since carbon was not transferred to the atmosphere as requested by the protocol but to a great proportion left onsite for decay. Still, CanESM shows a very interesting behaviour that diverges from the other models: CanESM simulates a uniform global increase in NPP associated with the highly productive C4 grasses, especially in the tropics. The increase in NPP is accompanied by almost as strong heterotrophic respiration increases (as a consequence of

increased litter and soil carbon pools) resulting in net ecosystem carbon gains. GPP changes are, however, not obviously different with mainly less productive grasses everywhere but in Africa and south-east Asia, meaning that autotrophic respiration of grasses decreases more than in all other models after deforestation.

The net biome productivity ($\Delta NBP = \Delta_t cLand$ with $\Delta_t$ referring to the year-to-year changes in cLand) summarizes the effect of

land carbon fluxes. In that, all models show similar carbon fluxes of -4.45±1.07 GtC yr$^{-1}$ during deforestation except for CESM2 with dominating influences from ΔfFire. After the end of deforestation, the ΔNBP reduction declines on average to -0.45±1.06 GtC yr$^{-1}$. The outliers MPI (continuous reductions of rh) and UKESM (recovering ΔNPP) show a positive trend thus initiating a slowdown of the loss in ΔcLand opposed to MIROC in which Δrh increases with the growth of secondary vegetation.

Land carbon changes emerge as a signal within the first 10 years in most places (Fig. S18). MPI and IPSL show more distinct patterns at the outermost edges of deforestation where 30 years pass before ToE occurs. In IPSL and CESM2, many patches outside ΔF show ToE of up to 50 years; however, changes are smaller than ±0.5 kg m$^{-2}$ in these areas. ToE of ΔGPP (Fig. S19) is more interesting. ΔGPP is uniformly affected by the replacement of trees by grasses but influenced also by the changes in local climate. In MPI and IPSL, the earliest ToE values appear where strong GPP reductions are observed, while in CESM2 these locations

experience strong GPP increases in this time (10-40 years). Changes in climate impose their signal and thus, similar patterns of propagation can be observed as for ToE of ΔTas.

Although forest removal was implemented in a similar way across models, the trajectory and spatial patterns of carbon changes differ strongly. The major part of the land carbon model spread stems from the removal of vegetation carbon based on the

differences in initial forest distribution and carbon densities. A similar divergence across multiple models but of lower magnitude was already found for a previous study investigating the effect of future land use and land cover changes on the carbon cycle in CMIP5 models (Brovkin et al., 2013). The changes in ΔcLand (-260±74 GtC) in the *deforest-globe* simulation are ~25% higher than the estimated historical emissions of 205±60 GtC from land-use and land cover changes and wood harvest and wood products between 1850 and 2018 (Friedlingstein et al., 2019).






The loss of land carbon follows a similar trajectory at the global scale with only vegetation recovery (MIROC, UKESM) and grass parametrization (CESM2) causing non-linearities. We find that not only whether fire is represented (MPI, CESM2, EC-Earth) or not can have substantial effects on the NBP and thus, overall carbon losses but especially, how it is implemented. In CESM2 fire is used as a deforestation tool, while it only depends on litter fluxes in MPI and EC-Earth. In the latter model, fire activity decreases

with the expansion of grassland opposed to the other two models. To narrow down the sign and magnitude of fire emissions thus needs further consolidation by incorporating observational data. The protocol allowed models to simulate dynamic vegetation processes outside the deforestation area based on the assumption that the time horizon of the experiment was too short for climate change effects to affect the remaining woody vegetation (Lawrence et al., 2016). UKESM disproves this assumption since forest cover continuously declined in the remaining part of the grid cell. The separation of land carbon pools by land cover type would

have therefore been advantageous. Across all models that witness a declining or constant fast soil carbon pool with the onset of deforestation (CESM2, IPSL, MIROC), the fate of below ground plant materials (roots) remains unclear considering that root biomass is about one-fifth of the above ground biomass (Lewis et al., 2019).

The analysis of CMIP5 models revealed that substantial uncertainty in model responses was due to implementation differences

(i.e., land use patterns, Boysen et al., 2014; Brovkin et al., 2013). Having a very simple experimental protocol of replacing trees with grasses, we now show that the underlying processes themselves also explain large parts of the model spread. Strong or weak model responses may originate from including or not representing certain processes explicitly e.g. fire activity or soil biochemistry. Our analyses also highlighted the relevance of the comparative response of different vegetation types. While most evaluation is done for total land carbon stocks and fluxes, assessment of land-use change requires adequate representation of individual land

use/cover types at each location relative to each other. This highlights the need for improving the process understanding of soil carbon dynamics (e.g. Chen et al., 2015; Don et al., 2011; Giardina et al., 2014; Luo et al., 2017), fluxes (Atkin et al., 2015; Huntingford et al., 2017) and biomass carbon stocks (Erb et al., 2017) using observations and field experiments.

### 3.3.2 Forest sensitivity (FS) of land carbon:

Similarly to ΔTas, the FS of ΔcLand is analysed. Across models, MIROC is the most sensitive with local land carbon losses per fraction of deforestation (Fig. S20) of ~-95 GtC, followed by UKESM with ~-60 GtC and other models with -40 to -50 GtC. On average, UKESM, CESM2 and MPI amount to -16 GtC while the other models stay around -9 GtC per fraction of deforestation. Per $10^3$km$^2$ of deforestation within one grid cell (Fig. S21), EC-Earth removes on average -2.7 GtC, CESM2 and UKESM around -1 GtC while the other models stay above -0.5 GtC.


The sensitivity of land carbon changes with regard to the deforestation fraction in a grid cell across latitudinal climate zones (Fig. S25) depends on the initial biomass carbon, soil carbon dynamics, the characteristics of the replacing vegetation and probably even climate.

Most models, except for MIROC, and IPSL, show an almost linear decrease of FS in the boreal and temperate region although the

magnitudes vary strongly (Fig. S22 and S23). On average, models decrease cLand by 4.1 and 4.9 kg m$^{-2}$ frac$^{-1}$ in the boreal and temperate zone, respectively. In the tropics, IPSL, CNRM (to a lesser degree MPI and UKESM) simulate on average weaker decreases around 30% deforestation than with lower and especially larger forest removals. These models including EC-Earth remove very similar amounts of carbon per deforestation amount. However, in EC-Earth, cLand changes in the tropical region almost stay constant above 60% deforestation. CESM2 simulates a strong negative non-linear behaviour (ca. -9 kg m$^{-2}$ frac$^{-1}$)





dominated by vegetation carbon removal in South America. MIROC reveals an increasing nonlinearity the further north the forest is removed. On average, the tropical cLand loss is quantified with 5.1 kg m$^{-2}$ frac$^{-1}$.

Although climatic changes affect the carbon cycle negatively via droughts or positively via favourable warming (see also Fig 9), the main contribution comes from the deforestation itself as also the temporal analysis revealed. Therefore, the carbon response to

ΔF is mainly local and almost linear. The FS approach can well be used to analyse the effects on land carbon pools in future land use scenarios to derive gross $CO_2$ emissions. The mechanisms behind FS of ΔcLand may differ across models and non-linear dynamics from vegetation distribution changes at the local scale can influence the results. For models like MIROC and BCC, we would not apply this approach as clearly drivers from climate or parametrization play a role. Also, the effects of changing atmospheric $CO_2$ concentrations on the carbon cycle are not captured in this study.

**3.3.3 Estimated changes in temperature due to ΔcLand**

Carbon emissions from deforestation in the real-world act as a greenhouse gas with a potential warming effect. In absence of varying $CO_2$ concentrations in this experimental setup, we can therefore only approximate the temperature effects of deforestation. The TCRE serves as a good tool to estimate the biogeochemical (BGC) effect on climate from large-scale deforestation (ΔTas in regard to ΔcLand). While the overall biogeophysically (BGP) induced effect of deforestation was a cooling on both the global

scale and also over most areas of ΔF, the BGC effect results in an estimated global warming of 0.18 to 0.85 °C. For MIROC, IPSL and MPI (0.18 to 0.57 °C) this is the dominant temperature response to ΔF assuming that the TCRE concept allows to calculate a significant temperature change from any statistically significant change in land carbon pools. The remaining models would yield a warming between 0.36 to 0.85 °C. Note that the result for CanESM and BCC is only based on cVeg changes. On the global scale, the BGC warming is at least similarly strong (CNRM, UKESM) or four times larger (CESM2) than the BGP cooling. When

considering areas of deforestation alone, the robust BGP cooling dominates in CanESM, CNRM and UKESM.

The estimate of ΔTas in regard to ΔcLand depends not only on each model's TCRE but also on the sensitivity of ΔcLand in regard to ΔF (see section 3.3.2). Thus, models with strong carbon losses (e.g. MPI) may still have lower climate sensitivities (e.g., 1.6°C EgC$^{-1}$) and vice versa, leading to a similar range of results. Although TCRE was shown to be a useful tool regardless of non-$CO_2$

and aerosol forcings, we here ignore the carbon-concentration feedback in the absence of variable $CO_2$ concentrations which could potentially enhance the land carbon sink via $CO_2$ fertilization (Bathiany et al., 2010). Nevertheless, while large-scale deforestation could lead to an overall BGP cooling globally and over ΔF, BGC warming dominates on the global scale with the possibility to balance boreal BGP cooling or enhance tropical BGP warming.



## 4. Conclusions

Nine Earth System models carried out the LUMIP global deforestation experiment (*deforest-glob*) of replacing 20 Mkm$^2$ forest with grassland over 50 years followed by a stabilization period of 30 years. The setup was designed to guarantee as much similarity in implementing a deforestation experiment across models as possible. Nevertheless, model structures differ leading to varying initial forest covers and thus, somewhat different patterns of deforestation.

The biogeophysical effect on mean global near-surface temperatures (ΔTas) across all models is a cooling of -0.41±0.41°C over areas of deforestation (ΔF) and -0.22±0.2°C globally. Non-local effects due to strong Arctic feedbacks (CanESM, CNRM, CESM2 and UKESM) induce this globally dominant cooling which also prevails over ΔF. Regionally, non-local effects may be caused by advection of air (e.g., BCC, MPI, IPSL in the tropics) across grid cells. The biogeophysical effects continue to grow through the entire simulation, with most models not having reached a new climate equilibrium by the end of the 30-year stabilization period.


While the biogeochemical effects of large-scale deforestation on total land carbon (ΔcLand) are consistent across most models (mean -269±80 GtC), the contributing fluxes and impacts on specific carbon pools differs strongly across models and regions particularly because the interplay of vegetation cover, carbon pools, moisture cycling and climate can be substantially different across models. The estimated temperature effect of the released carbon is a warming of 0.46±0.22°C which dominates globally

over the biogeophysically-induced cooling and enhances the tropical warming (except for UKESM and EC-Earth). Note, that possible negative or positive carbon-concentration feedbacks (i.e., $CO_2$-fertilization) are not accounted for in the model configurations used for these simulations.

Nonlinear responses with time underline the importance of accounting for amplifying non-local effects, showing for example that

the changes in temperature or GPP propagate from the centre to the edges of deforestation in the tropics. The detection of robust climate signals may take decades or require more than 30-50 % of a grid cell's forest cover to be removed - a very long time (or large area affected) for climate policies to act. Though these results were found to be causally plausible, they have to be treated with caution due to lack of a sufficiently large ensemble.

The *deforest-globe* simulations are useful also for generating look-up tables or deforestation-climate-emulators to provide quick and cheap analysis tools for deforestation scenarios. The new concept of ´forest sensitivity´ allows to derive good approximations for changes in temperature and atmospheric $CO_2$ from changes in land carbon, if the underlying processes are well understood. However, in case of strong climate feedbacks such as found in CanESM or UKESM, this application is limited to areas where non-local effects are small or not superimposed (e.g., tropics). A more detailed analysis on ecoregion and plant functional type level

would be necessary to guarantee a good level of representation.

Biogeophysical and biogeochemical model responses differ due to the varying characteristics of the replacing grass (CESM2, EC-Earth) or regrowing vegetation (MIROC and UKESM), soil parameters and dynamics and altered land-atmosphere coupling responsible, for example, the partitioning of available energy into turbulent fluxes or the moisture transfer. Not only the distribution

of initial forest cover but also the inherent carbon stocks differ widely and thus, their losses. Soil carbon and physiological dynamics of trees versus grasses and their dependence on climate need better understanding through incorporating observations and field studies to constrain fluxes like heterotrophic respiration, GPP or autotrophic plant respiration.



Future analyses of the *deforest-globe* experiments could further focus on the seasonality of vegetation and climate variables (e.g.
with regard to large scale atmospheric circulation changes) to advance the understanding compared to previous studies (Bonan, 2008; Chen and Dirmeyer, 2020; Davin and de Noblet-Ducoudré, 2010; de Noblet-Ducoudré et al., 2012; Pitman et al., 2009). Additionally, the enhanced meridional temperature gradient may alter the large-scale circulation that deserves future explorations. The comparison with observational data could refine the non-local responses (Duveiller et al., 2018a) across this wide range of models.


This study provides the first unified multi-model comparison of large-scale deforestation effects on climate and the carbon cycle. By reducing uncertainties from the land cover change implementation itself we showed that the remaining model spread largely stems from model parameterizations and process representation of trees and grasses which could be improved by incorporating observational data.


**Data availability**

Primary data and scripts used in the analysis and other supplementary information that may be useful in reproducing the author's work are archived by the Max Planck Institute for Meteorology and can be obtained <URL upon acceptance>.

**Supplement link**

(given by journal)

**Author contribution**

LB designed the study, performed all analysis including scripting and plotting and wrote the manuscript. VB wrote the introduction and the abstract together with LB. All co-authors contributed with editing the manuscript and giving suggestions on the analyses.

**Competing interests**

The authors declare that they have no conflict of interest.

**Acknowledgements**

LB, VB and JP thank Thomas Raddatz and Veronika Gayler for setting up the deforestation maps used in MPI and for managing the publication of the simulations. LB and VB acknowledge funding from the H2020 CRESCENDO project (grant agreement No
641816). LB and JP were funded by the DFG priority program SPP 1689 CE-Land+.
MR, CD and RS thank the CRESCENDO project under the grant agreement No 641816.
NV and PP acknowledge the HPC resources of TGCC under the allocations 2016-A0030107732, 2017-R0040110492 and 2018-R0040110492 (project gencmip6) provided by GENCI (Grand́Equipement National de Calcul Intensif) to conduct CMIP6 projects at IPSL.
LN acknowledges financial support from the Strategic Research Area MERGE (Modeling the Regional and Global Earth System - www.merge.lu.se), and from the Lund University Centre for Climate and Carbon Cycle Studies (LUCCI). PA acknowledges funding from the Helmholtz Association in its ATMO programme. MML acknowledges postdoctoral funding support from the James S. McDonnell Foundation.



EC simulations were performed on the Tetralith supercomputer of the Swedish National Infrastructure for Computing (SNIC) at
Linköping University under project SNIC 2018/2-11 (S-CMIP) and data handling was facilitated under project SNIC 2019/12-18
(SWESTORE), both partially funded by the Swedish Research Council through grant agreement no. 2016-07213. Simulations
were partially funded by the European Commission's Horizon 2020 Framework Programme, under Grant Agreement number
641816, the "Coordinated Research in Earth Systems and Climate: Experiments, kNowledge, Dissemination and Outreach
(CRESCENDO)"

The CESM project is supported primarily by the National Science Foundation (NSF). This material is based upon work supported
by the National Center for Atmospheric Research, which is a major facility sponsored by the NSF under Cooperative Agreement
No. 1852977. Computing and data storage resources, including the Cheyenne supercomputer (doi:10.5065/D6RX99HX), were
provided by the Computational and Information Systems Laboratory (CISL) at NCAR. DML was supported in part by the
RUBISCO Scientific Focus Area (SFA), which is sponsored by the Regional and Global Climate Modeling (RGCxM) Program in
the Climate and Environmental Sciences Division (CESD) of the Office of Biological and Environmental Research in the U.S.
Department of Energy Office of Science.

Min-Hui Lo is supported by the Ministry of Science and Technology in Taiwan under grant 106-2111-M-002-010-MY4

SL was supported by the CRESCENDO project and by Joint UK BEIS/Defra Met Office Hadley Centre Climate Programme
(GA01101). This project has received funding from the European Union's Horizon 2020 research and innovation programme under
grant agreement No 641816. SL thanks Eddy Robertson for creating the land use ancillaries and for helpful discussions.

 TH is supported by the "Integrated Research Program for Advancing Climate Models (TOUGOU program, JPMXD0717935715)"
from the Ministry of Education, Culture, Sports, Science and Technology (MEXT), Japan.



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
