# Peer review of "Global climate response to idealized deforestation in CMIP6 models"

_Biogeosciences, 2020_

## Referee Comment (RC1) · Anonymous Referee #1 · 28 Jul 2020

**Overview**

The manuscript by Boysen et al. quantifies the biogeophysical and biogeochemical effects of deforestation. A consistent configuration among Earth system models (ESMs) derives from the use of the "deforest-globe" experiment, which is part of the Land Use Model Intercomparison Project (LUMIP) effort. This analysis, then, presents a unique opportunity to evaluate the magnitude, pattern, and time of response to a global deforestation of about 20 million square kilometers. This analysis, then, gives an original and useful insight into the global and regional climate and carbon cycle response to a wide and rapid global deforestation. At this stage, the manuscript requires some clarifications before being ready for publication, as described in the following sections.

**General comments**

The analysis presented in this manuscript applies to nine Earth system models, and many variables are analyzed. For this reason, the correct reference and the proper variable-model linkage may be lost in the text. This issue, then, can make the text hard for the reader. Among the different "Results and Discussion" sections, section 3.3.1 results easier to follow. This section, indeed, contains a straightforward model-by-model description and variable analysis. In this case, then, it is easier to extrapolate information about each model and each analyzed variable. I think that a similar level of clarity should also be attained in the other "Results and Discussion" sections.

Moreover, six out of nine models present some divergence from the protocol. If the analysis is performed only on the remaining three models, is it changing the main results?

Among these six models, MIROC exhibits a limited response to the deforestation forcing. Since MIROC allows for a forest regrowth, and, so, a divergence from the protocol: is it worth to keep it in this analysis?

Specific comments on the unclear portions of the text are listed below.

**Specific comments**

Line 107: supplementary figure 2 is cited before figure S1. Invert figures S1 and S2.

Line 127: "kilometres" should be "kilometers".

Lines 150-161: Six out of nine models present some divergence from the protocol. If the analysis is performed only on the remaining three models, is it changing the main results?

Lines 163-164 and Line 206: How many members does MPI provide? Seven or eight?

Line 170: the "branching year" is not given for all models in Table S1.

Equation 1: Tsurf,piControl should be $T_{surf,piControl}$ with "surf,piControl" subscript.

Table 1: which symbol denotes non-significant changes (it is missing in my pdf version)?

Table 1: does "Pr" stand for precipitation?

Table 1: in "$\Delta$cLand over $\Delta$F" of "MPI" there is an extra parenthesis ")".

Figure 2 caption: "near surface air temperature" refers to $\Delta$Tas?

Figure 3: why don't you directly use Figure S3 as Figure 3? Figure S3, indeed, contains the same information in Figure 3, but it also shows the global versus deforested grid-cell comparison. In figure S3, you could add the multi-model mean values, as done in figure 5.

Lines 323-324: where can I see the changes in evapotranspiration and latent heat flux?

Line 326: "net shortwave", but the brown line in Figure 2 shows incoming shortwave and not the net one.

Line 327: the temperate and boreal areas are introduced, but the latitudinal range of these regions are only reported later on in the text (caption of Figure 4).

Line 328: Figure S5 shows the distribution of albedo changes. It is not showing that "increasing albedo are stronger than reduction in longwave radiation".

Lines 328-329: "net shortwave reduces", is it shown in figure S4c?

Line 329: "incoming shortwave radiation increases", is it brown line in Figure 3?

Lines 330-331: "MPI and IPSL reduction in cloud cover", is it shown in Figure S4f?

Line 332: "turbulent heat flux", is it shown in Figure S4a?

Line 342: "increase in albedo", is it visible in Figure S5?

Line 343: "reduced cloud cover", is it shown in Figure S6?

Line 343-344: where is shown the comparison between changes in incoming shortwave from clouds and incoming shortwave from albedo?

Line 347: "as also observed" should be "is also observed".

Line 352: "net warming", is it visible in Figure 2c?

Line 352: "evaporative cooling", is it shown in Figure S8i?

Line 354: "sensible heat fluxes", is it shown as cyan lines in Figure 3?

Line 355: "BCC temperature increase", is it visible in Figure 2f?

Line 359: "evaporation decrease", is it shown in Figure S8f?

Line 359: "increase cloud formation", the cloud increase seems limited over the Amazon in Figure S6f.

Line 361: "not shown", the difference between longwave and longwave clear sky is displayed in Figure S7. Are you referring to different variables? Otherwise, the difference is shown. The described difference is not visible to me in Figure S7.

Line 364: is CNRM case shown in Figure 3e?

Lines 369-370: formatting error: the new line is started too early.

Line 383: available energy should be Figure S4b instead of Figure S4a.

Line 385: "cloud formation", is it visible in Figure S4f?

Line 386: turbulent heat flux should be Figure S4a instead of Figure S4b.

Line 386: is latent heat flux visible in Figure S4d?

Line 388: "[…] over temperate and boreal over grassland […]", the second over should be deleted.

Lines 416-421: The temperature changes per unit fraction of grid cell deforested range between 4 and -20, why Figure S9 has values between 2 and -2?

Similarly, the temperature changes per unit area deforested reported in Figure S10 and the text cover a different range of values.

Line 431: "On average […] 0.27 °C frac$^{-1}$", does it refer to Figure 4a?

Line 434: "reverse at higher $\Delta F$", does it refer to Figure 4c?

Line 434: "regions_there" should be "regions (Fig. 4b) there".

Lines 465-467: "Over South America […] over Eurasia", where are these changes shown?

Line 485: "30% and 10% of deforestation still left", is it visible in Figure S12?

Line 506: "2013) and" should be "2013; and".

Line 572-573: "For all models but MIROC […] ($\Delta$cSoil).", is it visible in Figure 8?

Line 577-579: "Below ground carbon […] several decades)", are fast and medium soil carbon pools displayed in Figure S17?

Line 585: "large NEP", is it visible in Figure 10d?

Lines 594-595: are heterotrophic respiration and net ecosystem productivity shown in Figure 10?

Line 598: "NPP reduction", is it depicted in Figure 10b?

Lines 617-618: "cVeg change […] carbon pools", is it visible in Figure 8?

Lines 618-619: "The exception […] soil carbon pool", where is it shown?

Lines 621-622: "In BCC, […] simulation period", is it shown in Figure 10?

Line 628: "uniform global increase in NPP", is it visible in Figure 10b?

Lines 628-629: "especially in the tropics", where is it shown?

Lines 630-631: "GPP changes are […] south-east Asia", is it shown in Figure 9b?

Lines 634-636: NBP values are the ones shown in Figure 10f?

Lines 644-645: "GPP reductions […] GPP increase in time", are the changes shown in Figure 9?

Lines 680-684: The range of values displayed in Figures S20 and S21 are different from the values reported in the text. For example, Figure S20 shows values between -20 and 20 kg m-2 frac-1, while the text describes changes between -40 and -90 GtC.

Lines 686-687: "Fig. S25" should be "Fig. S22".

Lines 691-692: "In the tropics, IPSL […] forest removals.", is it visible in Figure S22a?

Line 695: "removal in South America", is it shown in Figure S20c?

Conclusions: the conclusions should also report the opposite biogeophysical response between boreal/temperate forests and tropical forests. The possibility to identify a "threshold" latitude below which re/afforestation could have both biogeophysical and biogeochemical cooling effect is significant.

Figure S2 caption: "Mkm2" should be "Mkm$^2$" with "2" as apex.

Figure S7: why doesn't this figure report CESM2?

Figure S9 caption: "frac-1" should be "frac$^{-1}$" with "-1" as apex.

Figure S10 caption: "km2" should be "$km^2$" with "2" as apex.

Figure S13: "Only statistically significant areas as found in Fig. 2 are shown." but the figure's legend report also "non-sign.". Are non-significant areas plotted or not?

Figure S15 caption: "m2" should be "$m^2$" with "2" as apex. Add why some models are not plotted.

Figure S16 caption: "km2 year-1" should be "$km^2$ $year^{-1}$" with "2" and "-1" as apex. Add why some models are not plotted.

Figure S18 caption: "ToE in years […]" should be "ToE and FoE in years […]".

Figure S19 caption: "Same as Fig S19 but for the ToE of gross primary productivity" should be "Same as Fig S18 but for the ToE and FoE of gross primary productivity".

Figure S20 caption: Add why some models are not plotted.

Figure S21 caption: Add why some models are not plotted.

Figure S23 caption: Add why some models are not plotted.

---

## Author Comment (AC1) · 10 Aug 2020

Dear referee,

thank you for your valuable feedback to our manuscript. In the following I will address your points and how we want to improve the manuscript.

General comments: We understand that it can sometimes be difficult to follow the model and variable analyses throughout the text. We will improve the readability further by better streamlining models according to their behavior with regard to observed phenomena and appreciate your suggestion. The reason behind the current structure is that for the particular result sections under 3.2. we had much more complex analyses to present than under 3.3.. The biogeophysical analysis required us to evaluate

many variables for different concepts e.g. surface energy balance decomposition, time of emergence or responses normalized by tree fraction. Analyzing the 'main variables of interest' such as near surface temperature or precipitation, we naturally had to draw the link to many other variables. This is easier achieved when analyzing carbon variables which are less numerous in the LUMIP protocol. We therefore structured around phenomena across models instead of the whole picture for each model separately as done in the carbon section. Apart from this pragmatic reason, we found the structuring around phenomena for biogeophysical effects helpful, because it better reveals common features across models, which in fact quite often exist for the climatic impacts.

We will add a paragraph on how results change if only the protocol-conform models are included.

We like to keep MIROC in the analyses. We account for the divergence to the protocol when discussing the results. Such weak responses are still an interesting model feature because forest regrowth does not happen instantly either.

Thanks for your close observations regarding the text readability and clarifications, linking figures and typos. We will work through them during the revision process.

With kind regards, Lena Boysen & co-authors

---

## Referee Comment (RC2) · David Lapola (Referee) · 18 Aug 2020

This is a relevant study that shows the modeled climatic impacts of an concerted removal of forests across the globe. The authors should be applauded for the effort in first conducting this multi-model exercise and also for the well-organized and systematized analysis. The article presents a proper level of self-critique regarding the lack of surface-atmosphere feedbacks such as the missing effects of CO2 emitted with such a deforestation and its physiological (e.g. CO2 fertilization) and radiative feedbacks to the land vegetation (this much should be highlighted in the abstract!).

My major concern is regarding the influence of the ocean on the resulting climatic patterns. In the introduction (L70) the authors state that atmosphere-only simulations of

this kind tend to present a stronger (-than-real?) signal-to-noise ratio for deforestation. Using ESMs is indeed a huge advantage over an atmosphere-only approach. However the authors lose an ideal opportunity to demonstrate this by showing the incurred changes in sea surface temperature and circulation patterns. There is historically a number of atmosphere-only modeling studies showing indeed a strong temperature sensitivity to deforestation, especially in the tropics. Not much has been said in these previous studies on the Time of Emergence, a nice tool the authors use in this current manuscript. Is the ToE presented here related to the climatic inertia of the ocean? The exploration of the ocean-related variables could clarify this, as well as other issues presented in the article (e.g. cooling of western Europe in some models and reduction of precipitation in the tropics).

As a minor suggestion, the authors could at least discuss the probable results from a separate experiment of deforestation in the boreal zone alone and another one of deforestation in the tropics alone.

Other minor comments:

L51: future losses in what time horizon?

L52: not only to free land for food and bioenergy but to speculate on land ownership itself (i.e. to justify land grabbing [see Rajão et al. 2020 Science]).

L70: "An experimental setup with atmosphere-only models in which sea surface temperatures are prescribed allows to increase the signal-to-noise ratio of models' response to deforestation. This setup assumes that the effect of large-scale circulation changes is small and can be ignored." Please explain this better. You mean the sensitivity of models to deforestation increases (artificially?) in atmosphere-only model setups? "(…)the effect of large-scale (atmospheric?) circulation changes is small (compared to what?) and can be ignored."

L125: if tree and forest are different things why using them interchangeably then?

Table 1: are you sure the units of delta Pr is mm/yr?

L323: is this somewhat related to an acceleration of the Atlantic THC? By the way that is the only region the IPCC shows as cooling in the SRES scenarios. Please explore the implications of this result deeper.

L353: the increase in leaf area is such that LAI surpasses that of a typical tropical forest?

L359: But this also decreases incoming shortwave radiation, doesn't it? Fig 3f doesn't show a considerable increase in longwave radiation.

L378: a "global" cooling effect due to CO2 removal...

L379: uncertain

L389-390: Is this based on in-field experimental studies or modeling? How trustful is it?

L414: Tas or Tsurf?

Fig 6a: ToE in the tropics seems restricted to 30 years or more (i.e. values smaller than 30yr do not occur). Is that due to the use of a 30-yr moving average?

L501: "...robustly detectable after a few decades". What is that time lag attributable to? Influence of slower thermal sensitivity of the adjacent oceans? That's one strong reason to show sea surface temperature and circulation changes in this article.

L508-509: but notice that this study did not test that (i.e. the climatic effects of re/afforestation). Would be a nice follow-up.

L523: What is an unfavourable climatic condition? I suggest a different term also on L546.

Fig 9: after how many years?

L591: I think the authors should present arguments why the C4 grasses are "overly"

productive. C4 grasses like sugarcane or others can indeed have a NPP higher than that of tropical forests.

L621: It is unclear why figure 9f shows a steep decline of GPP in northern Amazon, considering that considerable increase in precipitation for that specific region was previously presented in the paper for BCC.

L765: Why wasn't this analysis of large-scale circulation patterns done in this study?

L67: "...temperature gradient mau alter the large-scale circulation..." Atmospheric or oceanic?

L774: Is it possible to be a bit more specific on the sort of observational data needed?

Fig S13: the brownish color in the tropics in panel f and in the boreal zone in panel d doesn't seem to correspond to any color in the legend.

---

## Referee Comment (RC3) · Anonymous Referee #3 · 20 Aug 2020

Review of bg-2020-29:

This study reports on the multi-model LUMIP deforestation experiment. The authors show that controlled, large-scale, global deforestation may contribute global geophysical cooling of near-surface temperatures and global geochemical warming. The geophysical effects vary latitudinally and by model, generally with warming in the tropics and cooling elsewhere, while the geochemical effects are estimated offline as warming everywhere. The geochemical effects generally are greater than the geophysical effects, leading to net warming, although a potential $CO_2$-enhanced land sink is not included here. Land carbon losses are driven by vegetation loss. Some novel metrics for assessing and potentially estimating the effects of deforestation are also presented.

I appreciate the tremendous effort the authors have put into this study to advance our

understanding of the effects of land cover change on the earth system. I have a few main comments, followed by some brief detailed comments.

1) What are the take home messages? There are a lot of different analyses, and only one aspect is highlighted in the abstract. The abstract includes some key numbers, and at least the potential net warming statement of large scale deforestation. While this in itself is a key finding (with the associated caveat of constant $CO_2$), there are a few other notable results to highlight. Missing are the policy/scenario implications related to time/fraction of emergence for climate vs carbon. And the potential for rough estimates of response via the sensitivity metrics (which are analogous to climate sensitivity of models). I understand that there are some limitations to the sensitivity metrics and the time of emergence estimates, but based on S13 and S18 (plus the rest of the carbon figures), it seems safe to say that the climate signals have relatively long time frames while carbon signals have relatively short time frames. However, the climate signal emergence is further complicated by observations that show large, immediate meteorological distinctions between forest and grassland. The sensitivity metrics support the temperature and carbon results, and are potentially useful to the community.

2) In relation to comment (1) above, switching some of the regular and supplemental figures would make the paper stronger. For example, figures S22 (carbon sensitivity to deforestation) and S18 (ToE and FoE) are more relevant to the carbon points than figures 9-10, which are explanatory. Also, figure S3 is much clearer and easier to understand than figure 3, and follows the text better (you can add Tas to fig S3).

3) The descriptions of Tsurf-model and Tas are not complete, which makes it difficult to properly assess the temperature comparisons. While Tsurf is clearly a radiative temperature, is Tsurf-model a radiative temperature for all the models? Tsurf in some models is a canopy air temperature, at a height dictated by the displacement height and aerodynamic roughness. The 2m air temperature is often the air temperature 2m above this "Tsurf." It is important to be clear as to what and where these temperatures actually are, as shown by some of your references.

4) MIROC does not seem to meet the deforestation harmonization requirements, and its plots don't seem to add to the understanding of the issue. In fact, a lot of extra text is dedicated to explaining why MIROC is different from the others. It would be cleaner if it were not included.

5) Some supplemental figures are cited out of order.

With respect to the effects of differences in initial forest cover on the implementation of deforestation (lines 229-227), the author's may be interested in this recently published paper:

A.V. Di Vittorio, X. Shi, B. Bond-Lamberty, K. Calvin, A. Jones, 2020, "Initial land use/cover distribution substantially affects global carbon and local temperature projections in the integrated Earth system model", Global Biogeochemical Cycles. doi: 10.1029/2019GB006383.

Specific comments:

lines 326-327: this isn't clear from fig 3. fig s4 is more appropriate here.

line 465: which figures show these regional effects?

line 505: relate toe and implications to observation of immediate temperature differences between forest and grassland, and perceived differences

line 536: this paragraph is out of place - it doesn't relate to the rest of the section

line 644-646: fig S19 shows declines in GPP for CESM throughout the deforestation area, so it isn't clear how CESM has increases in GPP where the other model have decreases.

lines 680-684: based on fig S20, it doesn't appear the miroc can have the highest sensitiviy. most of its coverage has the smallest change in c per fraction of deforestation.

lines 710-713: this is unclear - you have separated your total range in two, arbitrarily,

and included veg only change models in one group with other total land c change models.

---

## Author Comment (AC2) · 7 Sep 2020

Dear Referee,

**Thanks for your close observations regarding the text readability and clarifications, linking figures and typos. We will work through them during the revision process (See a detailed list below).**

**With kind regards,**
**Lena Boysen & co-authors**

Line 107: supplementary figure 2 is cited before figure S1. Invert figures S1 and S2.
**Reply**: We will change the order of the figures in the SI.
Line 127: "kilometres" should be "kilometers".
**Reply**:  We will correct this.
Lines 150-161: Six out of nine models present some divergence from the protocol. If the analysis is performed only on the remaining three models, is it changing the main results?
**Reply**: We will add the results for only the three models that correctly followed the protocol.
Line 163: How many members does MPI provide? Seven or eight?
**Reply**: MPI provided seven members; we will correct this in line 206.
Line 170: the "branching year" is not given for all models in Table S1.
**Reply:** We will add the branching year for all models.
Equation 1: Tsurf,piControl should be $T_{surf,piControl}$ with "surf,piControl" subscript.
**Reply:** We will correct the subscript.
Table 1: which symbol denotes non-significant changes (it is missing in my pdf version)?
**Reply:** As stated in the caption, non-significant values are shown in parentheses.
Table 1: does "Pr" stand for precipitation?
**Reply:**  We will add that 'Pr' stands indeed for precipitation.
Table 1: in "DcLand over DF" of "MPI" there is an extra parenthesis ")".
**Reply:** We will delete the parenthesis around MPI.
Figure 2: "near surface air temperature" refers to DTas?
**Reply:** We will add '(ΔTas)' to the caption.
Figure 3: why don't you directly use Figure S3 as Figure 3? Figure S3, indeed, contains the same information in Figure 3, but it also shows the global versus deforested grid-cell comparison. In figure S3, you could add the multi-model mean values, as done in figure 5.
**Reply:** We will swap Fig. 3 and S3 as another referee also requested. We see the value in both figures which add different messages to the manuscript.
Lines 323-324: where can I see the changes in evapotranspiration and latent heat flux?
**Reply:** ET is shown in Fig. S8, latent heat fluxes are not displayed in a map. We will add the link and '(not shown)' to the text.
Line 326: "net shortwave", but the brown line in Figure 2 shows incoming shortwave and not the net one.
**Reply**: We will add that this can better be seen in Fig. S4c.
Line 327: the temperate and boreal areas are introduced, but the latitudinal range of these regions are only reported later on in the text (caption of Figure 4).
**Reply:** We will add the latitudinal range.
Line 328: Figure S5 shows the distribution of albedo changes. It is not showing that "increasing albedo are stronger than reduction in longwave radiation".
**Reply:** We will shift 'Fig S5' to right behind the mentioning of 'albedo'.
Lines 328-329: "net shortwave reduces", is it shown in figure S4c?

**Reply:** We will add '(Fig. S4c)'.

Line 329: "incoming shortwave radiation increases", is it brown line in Figure 3?

**Reply:** Yes, it is. We will add this.

Lines 330-331: "MPI and IPSL reduction in cloud cover", is it shown in Figure S4f?

**Reply:** Yes, we will add the link.

Line 332: "turbulent heat flux", is it shown in Figure S4a?

**Reply:** Yes, we will add the link.

Line 342: "increase in albedo", is it visible in Figure S5?

**Reply:** This should be 'increase in the albedo… (Fig. 5)'.

Line 343: "reduced cloud cover", is it shown in Figure S6?

**Reply:** Yes, we will add the link.

Line 343-344: where is shown the comparison between changes in incoming shortwave from clouds and incoming shortwave from albedo?

**Reply:** This was inferred by comparing the contributions of albedo and incoming shortwave radiation itself on Tsurf. For IPSL, CNRM, Can and BCC, the incoming shortwave radiation contributes with a greater increase in Tsurf than the reduction caused by albedo does. We will add the references to the text.

Line 347: "as also observed" should be "is also observed".

**Reply**: This will be corrected.

Line 352: "net warming", is it visible in Figure 2c?

**Reply:** Yes, and Fig. 3c. We will add the link.

Line 352: "evaporative cooling", is it shown in Figure S8i?

**Reply:** Yes, and Fig. S3i. We will add the links.

Line 354: "sensible heat fluxes", is it shown as cyan lines in Figure 3?

**Reply:** Especially in Fig S3d and S4e. We will add the link.

Line 355: "BCC temperature increase", is it visible in Figure 2f?

**Reply:** Yes, and S3f. We will add the link.

Line 359: "evaporation decrease", is it shown in Figure S8f?

**Reply:** Yes! We will add the link.

Line 359: "increase cloud formation", the cloud increase seems limited over the Amazon in Figure S6f.

**Reply:** Yes, but the text passage only deals with this region, so this should be covered.

Line 361: "not shown", the difference between longwave and longwave clear sky is displayed in Figure S7. Are you referring to different variables? Otherwise, the difference is shown. The described difference is not visible to me in Figure S7.

**Reply:** Indeed, this is shown in Fig. S7 but hard to detect. This is a very local region in northern Amazon but which leads to a strong temperature signal.

Line 364: is CNRM case shown in Figure 3e?

**Reply:** Yes, we will add the link.

Lines 369-370: formatting error: the new line is started too early.

**Reply**: This will be corrected.

Line 383: available energy should be Figure S4b instead of Figure S4a.

**Reply:** Yes, indeed.

Line 385: "cloud formation", is it visible in Figure S4f?

**Reply:** Yes, we will add the link.

Line 386: turbulent heat flux should be Figure S4a instead of Figure S4b.

**Reply:** Yes, indeed.

Line 386: is latent heat flux visible in Figure S4d?

**Reply**: Yes, and S8.

Line 388: "[...] over temperate and boreal over grassland [...]", the second over should be deleted.

**Reply:** Will be corrected.

Lines 416-421: The temperature changes per unit fraction of grid cell deforested range between 4 and -20, why Figure S9 has values between 2 and -2?

Similarly, the temperature changes per unit area deforested reported in Figure S10 and the text cover a different range of values.

**Reply:** Only a few pixels show these extreme values; If we extended the color bar range we would lose the visibility of the main range. We will add '(Note that the color bar range is limited and extreme values not shown).'

Line 431: "On average [...] 0.27 °C frac$^{-1}$", does it refer to Figure 4a?

**Reply:** Yes, we will add '(derived from Fig. 4a)'.

Line 434: "reverse at higher DF", does it refer to Figure 4c?

**Reply:** Yes, we will add the link.

Line 434: "regions_there" should be "regions (Fig. 4b) there".

**Reply**: This will be corrected and the link added.

Lines 465-467: "Over South America [...] over Eurasia", where are these changes shown?

**Reply:** The regional time series is not shown. We have added 'not shown' to the text.

Line 485: "30% and 10% of deforestation still left", is it visible in Figure S12?

**Reply:** Yes, we will add the link.

Line 506: "2013) and" should be "2013; and".

**Reply:** This will be corrected.

Lines 572-573: "For all models but MIROC [...] (DcSoil).", is it visible in Figure 8?

**Reply:** Yes, we will add the link.

Lines 577-579: "Below ground carbon [...] several decades)", are fast and medium soil carbon pools displayed in Figure S17?

**Reply:** Yes, we will add the link.

Line 585: "large NEP", is it visible in Figure 10d?

**Reply:** Yes, we will add the link.

Lines 594-595: are heterotrophic respiration and net ecosystem productivity shown in Figure 10?

**Reply**: Yes, we will add the links.

Line 598: "NPP reduction", is it depicted in Figure 10b?

**Reply:** Yes, we will add the link.

Lines 617-618: "cVeg change [...] carbon pools", is it visible in Figure 8?

**Reply:** Regional soil carbon time series are not shown. We will add this to the text.

Lines 618-619: "The exception [...] soil carbon pool", where is it shown?

**Reply:** We will add '(not shown)'. Listing all regional time series would be too much for the SI.

Lines 621-622: "In BCC, [...] simulation period", is it shown in Figure 10?

**Reply:** Yes, we will add the link.

Line 628: "uniform global increase in NPP", is it visible in Figure 10b?

**Reply:** Yes, we will add the link.

Lines 628-629: "especially in the tropics", where is it shown?

**Reply:** In Fig. 9d. We will add the link.

Lines 630-631: "GPP changes are [...] south-east Asia", is it shown in Figure 9b?

**Reply:** In Fig. 9d. We will add the link.

Lines 634-636: NBP values are the ones shown in Figure 10f?

**Reply**: Yes, we will add the link.

Lines 644-645: "GPP reductions [...] GPP increase in time", are the changes shown in Figure 9?

**Reply:** Yes, we will add the link.

Lines 680-684: The range of values displayed in Figures S20 and S21 are different from the values reported in the text. For example, Figure S20 shows values between -20 and 20 kg m-2 frac-1, while the text describes changes between -40 and -90 GtC.

**Reply:** The range displayed in the color bar is not the complete range. Adding the extreme values would lead to less visibility of the main range.

Lines 686-687: "Fig. S25" should be "Fig. S22".

**Reply**: This will be corrected.

Lines 691-692: "In the tropics, IPSL [...] forest removals.", is it visible in Figure S22a?

**Reply:** Yes, we will add the link.

Line 695: "removal in South America", is it shown in Figure S20c?

**Reply**: It's S23c. We will add the link.

**Conclusions**: the conclusions should also report the opposite biogeophysical response between boreal/temperate forests and tropical forests. The possibility to identify a "threshold" latitude below which re/afforestation could have both biogeophysical and biogeochemical cooling effect is significant.

**Reply**: We will add 'On average, the switch of sign from tropical warming to extra-tropical cooling happens around 22.6°N, thus just outside the tropics.'.

Figure S2: caption: "Mkm2" should be "Mkm$^2$" with "2" as apex.

**Reply**: This was an issue of converting the .docx to PDF. It will be fixed.

Figure S7: why doesn't this figure report CESM2?

**Reply:** CESM2 did not provide clear-sky longwave radiation. We will add this.

Figure S9: caption: "frac-1" should be "frac$^{-1}$" with "-1" as apex.

**Reply**: This was an issue of converting the .docx to PDF. It is fixed now.

Figure S10: caption: "km2" should be "km$^2$" with "2" as apex.

**Reply:** This was an issue of converting the .docx to PDF. It will be fixed.

Figure S13: "Only statistically significant areas as found in Fig. 2 are shown." but the figure's legend report also "non-sign.". Are non-significant areas plotted or not?

**Reply**: True, we only display significant values. We will correct the legend.

Figure S15: caption: "m2" should be "m$^2$" with "2" as apex. Add why some models are not plotted.

**Reply:** This was an issue of converting the .docx to PDF. It will be fixed.

Figure S16: caption: "km2 year-1" should be "km$^2$ year$^{-1}$" with "2" and "-1" as apex. Add why some models are not plotted.

**Reply:** This was an issue of converting the .docx to PDF. It will be fixed.

Figure S18: caption: "ToE in years [...]" should be "ToE and FoE in years [...]".

**Reply:** It should actually be 'ToE in years and the corresponding FoE in %...'.

Figure S19: caption: "Same as Fig S19 but for the ToE of gross primary productivity" should be "Same as Fig S18 but for the ToE and FoE of gross primary productivity".

**Reply:** This will be corrected.

Figure 20-23: caption: Add why some models are not plotted.

**Reply:** We will add to Fig. S15, S20, S21 and S23: 'Note, that CanESM and BCC did not follow the protocol when removing forests (see section 2.2) and are therefore not displayed for land carbon pools.'

---

## Author Comment (AC3) · 7 Sep 2020

Dear David Lapola,

thank you for your acknowledgement of our work and your valuable feedback to our manuscript. In the following I will address your points and how we want to improve the manuscript.

Referee comment: This is a relevant study that shows the modeled climatic impacts of a concerted removal of forests across the globe. The authors should be applauded for the effort in first conducting this multi-model exercise and also for the well-organized and systematized analysis. The article presents a proper level of self-critique regarding the lack of surface-atmosphere feedbacks such as the missing effects of CO2 emitted

with such a deforestation and its physiological (e.g. CO2 fertilization) and radiative feedbacks to the land vegetation (this much should be highlighted in the abstract!).

Reply: Thank you for you for this positive comment and acknowledgement.

Referee comment: My major concern is regarding the influence of the ocean on the resulting climatic patterns. In the introduction (L70) the authors state that atmosphere-only simulations of this kind tend to present a stronger (-than-real?) signal-to-noise ratio for deforestation. Using ESMs is indeed a huge advantage over an atmosphere-only approach. How- ever the authors lose an ideal opportunity to demonstrate this by showing the incurred changes in sea surface temperature and circulation patterns. There is historically a number of atmosphere-only modeling studies showing indeed a strong temperature sensitivity to deforestation, especially in the tropics. Not much has been said in these previous studies on the Time of Emergence, a nice tool the authors use in this current manuscript. Is the ToE presented here related to the climatic inertia of the ocean? The exploration of the ocean-related variables could clarify this, as well as other issues presented in the article (e.g. cooling of western Europe in some models and reduction of precipitation in the tropics).

Reply concerning the influence of the ocean:

We will discuss the role of the ocean a bit more. However, we tend to not include new variables or foci to the study and would leave such detailed analysis to follow-up studies (e.g. as currently done for the changes in atmospheric circulation). A thorough analysis of ocean dynamics would also require longer simulations that are currently not available to assess changes to the ocean stratification, mixing etc. As shown in Figure 2 (dTas) oceans show hardly any response except for models with strong sea-ice-albedo feedback (CanESM, UKESM) or those simulating the 'warming blob' (IPSL, CESM2, UKESM). Attached, you can find the plot for surface temperature which resembles the sea surface temperature over ice-free oceans. Over oceans, there are only little differences to detect compared to Figure 2. Consequently, changes in latent or sensible

heat fluxes (also attached) are not very pronounced. We will however include the below ocean-related explanations and discussions in the text. Thanks for pointing them out!

Reply concerning the warming blob caused by accelerated THC? (Line 323):

Yes, the acceleration of the THC could cause the observed "warming blob". This result would be in line with the opposed finding of Rahmstorf et al. (2015) who found a cooling blob due to freshwater input caused by global warming. We will add this to the manuscript.

Reply regarding why atmosphere-only simulations overestimate the signal-to-noise ratio (Line 70):

Davin and de Noblet-Decoudré (2010) separated the effects of global deforestation on global near-surface temperature with and without coupled ocean. The main component involved is the decreased land surface albedo. Less radiation is absorbed by the surface, thus the whole troposphere is cooled and contains less moisture. This effect is then transferred to the ocean surface which receives less longwave radiation from the atmosphere. Prescribing the ocean SST would lead to an overestimated tropical warming and underestimated boreal cooling. We will explain this more clearly in the manuscript.

Reply concerning the inertia of ToE signals (Line 501):

In order to show a detectable signal, dTas has to be twice as large as the climate variability. The ocean certainly plays a role in that it influences the variability and stabilizes climate through its inertia in heat uptake and feedback to the observed climate on land. Thus, the deforestation extent has to reach a certain magnitude. We will clarify this.

Reply concerning mentioning probable results from tropical or boreal deforestation alone:

The deforest-globe experiment was set up to cover both regions, the boreal and the tropical. Speculating on the effects of deforestation of only one region is very difficult,

especially, as models simulate varying responses. We would therefore like to avoid to do this.

Thank you for your further comments to the text which we will take into account when revising the manuscript and you can find them in the supplement.

With kind regards, Lena Boysen & co-authors

Please also note the supplement to this comment:
https://bg.copernicus.org/preprints/bg-2020-229/bg-2020-229-AC3-supplement.pdf

─────────────────────────

**Supplement:**

Dear David Lapola,

**thank you for your acknowledgement of our work and your valuable feedback to our manuscript. In the following I will address your points and how we want to improve the manuscript.**

This is a relevant study that shows the modeled climatic impacts of a concerted removal of forests across the globe. The authors should be applauded for the effort in first conducting this multi-model exercise and also for the well-organized and systematized analysis. The article presents a proper level of self-critique regarding the lack of surface-atmosphere feedbacks such as the missing effects of CO2 emitted with such a deforestation and its physiological (e.g. CO2 fertilization) and radiative feedbacks to the land vegetation (this much should be highlighted in the abstract!).

**Thank you for you for this positive comment and acknowledgement.**

My major concern is regarding the influence of the ocean on the resulting climatic patterns. In the introduction (L70) the authors state that atmosphere-only simulations of this kind tend to present a stronger (-than-real?) signal-to-noise ratio for deforestation. Using ESMs is indeed a huge advantage over an atmosphere-only approach. How- ever the authors lose an ideal opportunity to demonstrate this by showing the incurred changes in sea surface temperature and circulation patterns. There is historically a number of atmosphere-only modeling studies showing indeed a strong temperature sensitivity to deforestation, especially in the tropics. Not much has been said in these previous studies on the Time of Emergence, a nice tool the authors use in this current manuscript. Is the ToE presented here related to the climatic inertia of the ocean? The exploration of the ocean-related variables could clarify this, as well as other issues presented in the article (e.g. cooling of western Europe in some models and reduction of precipitation in the tropics).

Influence of the ocean:
**We will discuss the role of the ocean a bit more. However, we tend to not include new variables or foci to the study and would leave such detailed analysis to follow-up studies (e.g. as currently done for the changes in atmospheric circulation). A thorough analysis of ocean dynamics would also require longer simulations that are currently not available to assess changes to the ocean stratification, mixing etc.**
**As shown in Figure 2 (dTas) oceans show hardly any response except for models with strong sea-ice-albedo feedback (CanESM, UKESM) or those simulating the 'warming blob' (IPSL, CESM2, UKESM). Attached, you can find the plot for surface temperature which resembles the sea surface temperature over ice-free oceans. Over oceans, there are only little differences to detect compared to Figure 2. Consequently, changes in latent or sensible heat fluxes (also attached) are not very pronounced.**
**We will however include the below ocean-related explanations and discussions in the text. Thanks for pointing them out!**

Warming blob caused by accelerated THC? (Line 323)
**Yes, the acceleration of the THC could cause the observed "warming blob". This result would be in line with the opposed finding of Rahmstorf et al. (2015) who found a cooling blob due to freshwater input caused by global warming. We will add this to the manuscript.**

Clarifying why atmosphere-only simulations overestimate the signal-to-noise ratio (Line 70):
**Davin and de Noblet-Decoudré (2010) separated the effects of global deforestation on global near-surface temperature with and without coupled ocean. The main component involved is the decreased land surface albedo. Less radiation is absorbed by the surface, thus the whole troposphere is cooled and contains less moisture. This effect is then transferred to the ocean surface which receives less longwave radiation from the atmosphere. Prescribing the ocean SST**

**would lead to an overestimated tropical warming and underestimated boreal cooling. We will explain this more clearly in the manuscript.**

Inertia of ToE signals (Line 501):
**In order to show a detectable signal, dTas has to be twice as large as the climate variability. The ocean certainly plays a role in that it influences the variability and stabilizes climate through its inertia in heat uptake and feedback to the observed climate on land. Thus, the deforestation extent has to reach a certain magnitude. We will clarify this.**

As a minor suggestion, the authors could at least discuss the probable results from a separate experiment of deforestation in the boreal zone alone and another one of deforestation in the tropics alone.
**Reply: The deforest-globe experiment was set up to cover both regions, the boreal and the tropical. Speculating on the effects of deforestation of only one region is very difficult, especially, as models simulate varying responses. We would therefore like to avoid to do this.**

**Thank you for your further comments to the text which we will take into account when revising the manuscript (see detailed list below).**

**With kind regards,**
**Lena Boysen & co-authors**

Line 51: future losses in what time horizon?
**Reply: We will add 'until the end of the century'.**

Line 52: not only to free land for food and bioenergy but to speculate on land ownership itself (i.e. to justify land grabbing [see Rajão et al. 2020 Science]).
**Reply: Yes, that is true. However, to our knowledge this is not included in Hurtt et al. (2020) and therefore we would like to leave it as it is.**

Line 70: "An experimental setup with atmosphere-only models in which sea surface tem- peratures are prescribed allows to increase the signal-to-noise ratio of models' re- sponse to deforestation. This setup assumes that the effect of large-scale circulation changes is small and can be ignored." Please explain this better. You mean the sen- sitivity of models to deforestation increases (artificially?) in atmosphere-only model setups? "(. . .) the effect of large-scale (atmospheric?) circulation changes is small (compared to what?) and can be ignored."
**Reply: We will add: 'The cooling of the of the land surface via enhanced albedo cools and dries the whole troposphere which in turn transfers this signal via reduced longwave radiation further to the ocean. With prescribed SSTs the mediating effect of the ocean on the land temperatures is missing resulting in overestimated tropical warming and underestimated boreal cooling (Davin and de Noblet-Ducoudré, 2010).'**

Line 125: if tree and forest are different things why using them interchangeably then?
**Reply: We will change all 'tree fractions' or 'tree area' to 'forest fractions' or 'forest area'. We deleted the line 'Note, that we here use the terms tree and forest fraction interchangeable, although they are defined distinctively in reality.'**

Table 1: are you sure the units of delta Pr is mm/yr?

**Reply: Yes.**

Line 323: is this somewhat related to an acceleration of the Atlantic THC? By the way that is the only region the IPCC shows as cooling in the SRES scenarios. Please explore the implications of this result deeper.
**Reply: We will add: 'This result is in line with the reversed finding by Rahmstorf et al. (2015) who found a 'cooling blob' due to the freshwater input from the Greenland ice shield caused by global warming slowing down the meridional overturning circulation.'**

Line 353: the increase in leaf area is such that LAI surpasses that of a typical tropical forest?
**Reply: Yes, since the regrowth of forest is suppressed in this setup.**

Line 359: But this also decreases incoming shortwave radiation, doesn't it? Fig 3f doesn't show a considerable increase in longwave radiation.
**Reply: This observed phenomena is very local and averaged out in Fig. 3. You would see it in a map of shortwave radiation which is not shown and you can actually see in in Fig. S6 and S7. We will add this to the text.**

Line 378: a "global" cooling effect due to CO2 removal. . .
**Reply: Will be added.**

Line 379: uncertain
**Reply: Will be corrected.**

Lines 389-390: Is this based on in-field experimental studies or modeling? How trustful is it?
**Reply: Winckler et al. (2019b) performed model simulations with the MPI-ESM but also compared the results with various observational data sets which capture by nature only local effects.**

Line 414: Tas or Tsurf?
**Reply: 'Tas'**

Fig. 6a: ToE in the tropics seems restricted to 30 years or more (i.e. values smaller than 30yr do not occur). Is that due to the use of a 30-yr moving average?
**Reply: If you zoom into IPSL or CESM2 you can recognize single pixels with brighter colors / years below 30. The data was processed with the same script (no moving average applied!) so we can conclude that no robust effects appear in MPI before 30 years.**

Line 501: ". . .robustly detectable after a few decades". What is that time lag attributable to? Influence of slower thermal sensitivity of the adjacent oceans? That's one strong reason to show sea surface temperature and circulation changes in this article.
**Reply: We will add '… decades as climate variability and mediating effects from the ocean have to be overcome (Davin and de Noblet-Ducoudré, 2010).'**

Lines 508-509: but notice that this study did not test that (i.e. the climatic effects of re/afforestation). Would be a nice follow-up.
**Reply: We will change 'are' to 'could be'.**

Line 523: What is an unfavourable climatic condition? I suggest a different term also on L546.
**Reply: We will add '(too dry or hot)'.**

Fig. 9: after how many years?

**Reply: Like all maps as described in the methods section 2.3 first sentence: Climatic variables cover the last 30 years, carbon variables the last 10 years. We will add 'averaged over year 50 to year 79' or 'averaged over year 70 to year 79' to every spatial plot.**

Line 591: I think the authors should present arguments why the C4 grasses are "overly" productive. C4 grasses like sugarcane or others can indeed have a NPP higher than that of tropical forests.
**Reply: C4 grasses are parametrized in a way that they survive in arid regions. When entering the tropics, they are therefore too productive. This will be laid out in line 523 (see above).**

Line 621: It is unclear why figure 9f shows a steep decline of GPP in northern Amazon, considering that considerable increase in precipitation for that specific region was pre- viously presented in the paper for BCC.
**Reply: We will add 'Only in northern Amazon region, GPP is reduced under high temperatures and despite the observed precipitation increase (Fig. 7).'**

Line 765: Why wasn't this analysis of large-scale circulation patterns done in this study?
**Reply: Such a thorough study deserves its own publication and would exceed the scope for this manuscript (BGP and BGC effects).**

Line 767: ". . .temperature gradient mau alter the large-scale circulation. . ." Atmospheric or oceanic?
**Reply: We will add 'atmospheric'.**

Line 774: Is it possible to be a bit more specific on the sort of observational data needed?
**Reply: We have listed Duveiller et al. (2018b) who created a spatially and temporally explicit data set based on various observational data sets (ESA CCI and MODIS) to investigate the non-local effects to land cover change.**

Fig. S13: the brownish color in the tropics in panel f and in the boreal zone in panel d doesn't seem to correspond to any color in the legend.
**Reply: This will be fixed.**

---

## Author Comment (AC4) · 7 Sep 2020

Dear Referee,

thank you for providing us with this constructive review.

Referee comments: This study reports on the multi-model LUMIP deforestation experiment. The authors show that controlled, large-scale, global deforestation may contribute global geophysical cooling of near-surface temperatures and global geochemical warming. The geo- physical effects vary latitudinally and by model, generally with warming in the tropics and cooling elsewhere, while the geochemical effects are estimated offline as warming everywhere. The geochemical effects generally are greater than the geophysical effects, leading to net warming, although a potential CO2-

enhanced land sink is not included here. Land carbon losses are driven by vegetation loss. Some novel metrics for assessing and potentially estimating the effects of deforestation are also presented. I appreciate the tremendous effort the authors have put into this study to advance our understanding of the effects of land cover change on the earth system. I have a few main comments, followed by some brief detailed comments.

Reply: Thank you for your acknowledgement of our work and your valuable feedback to our manuscript. In the following I will address your points and how we want to improve the manuscript.

1. Improving the abstract: What are the take home messages? There are a lot of different analyses, and only one aspect is highlighted in the abstract. The abstract includes some key numbers, and at least the potential net warming statement of large scale deforestation. While this in itself is a key finding (with the associated caveat of constant $CO_2$), there are a few other notable results to highlight. Missing are the policy/scenario implications related to time/fraction of emergence for climate vs carbon. And the potential for rough estimates of response via the sensitivity metrics (which are analogous to climate sensitivity of models). I understand that there are some limitations to the sensitivity metrics and the time of emergence estimates, but based on S13 and S18 (plus the rest of the carbon figures), it seems safe to say that the climate signals have relatively long time frames while carbon signals have relatively short time frames. However, the climate signal emergence is further complicated by observations that show large, immediate meteorological distinctions between forest and grassland. The sensitivity metrics sup- port the temperature and carbon results, and are potentially useful to the community.

Reply: Thank you for pointing this obvious finding out! We will follow your suggestions and make the abstract stronger and more significant by including these results.

2. Swap figures: In relation to comment (1) above, switching some of the regular and supplemental figures would make the paper stronger. For example, figures S22

(carbon sensitivity to deforestation) and S18 (ToE and FoE) are more relevant to the carbon points than figures 9-10, which are explanatory. Also, figure S3 is much clearer and easier to understand than figure 3, and follows the text better (you can add Tas to fig S3).

Reply: We agree that some of the figures have the potential to be shown in the main text. However, we would like to keep the number of figures as it is now and the decision for one or the other figure is difficult. We would like to keep the GPP and carbon time series figures in the main text as they show fundamental results. The decision for the surface energy balance decomposition figure has also more sides to it. We argue that the model-wise comparison in the manuscript offers an easier access to the model's performance to simulate Tsurf. On the other hand, a component-wise presentation provides a better inter-model comparison which might be more valuable to the reader. We therefore agree to swap Figures 3 and S3.

3. Temperature definitions: The descriptions of Tsurf-model and Tas are not complete, which makes it difficult to properly assess the temperature comparisons. While Tsurf is clearly a radiative temperature, is Tsurf-model a radiative temperature for all the models? Tsurf in some models is a canopy air temperature, at a height dictated by the displacement height and aerodynamic roughness. The 2m air temperature is often the air temperature 2m above this "Tsurf." It is important to be clear as to what and where these temperatures actually are, as shown by some of your references.

Reply: Thank you for this inquiry. We will add more specifications to the method and results section on where surface temperature is calculated in the models and what this implies.

4. MIROC: MIROC does not seem to meet the deforestation harmonization requirements, and its plots don't seem to add to the understanding of the issue. In fact, a lot of extra text is dedicated to explaining why MIROC is different from the others. It would be cleaner if it were not included.

Reply: We would like to keep MIROC in the analysis. We list the caveats of each model's execution of the experiment and highlight if this is the reason for an observed result. This in itself is a demonstration of the difficulties in carrying out harmonized land-use and land cover change-related studies with many participating models. We think that it is still interesting how strong regrowth in this model is and how small the biogeophysical responses are despite a clear disturbance. Furthermore, MIROC simulates interesting carbon dynamics and should therefore be accounted for. We argue that it would be more inclusive to leave MIROC in all analyses than to keep it in only some relevant ones (e.g. the carbon analysis).

We will add 'We nevertheless analyse results from MIROC to not only demonstrate the effect these different technical realizations of one scenario can have but to also to draw conclusions for improvements in this model.'

5) Some supplemental figures are cited out of order.

Reply: We will fix that. Thank you for your close observation.

Thank you for your further comments to improve the readability and understanding of our study. We will take them into account during the revision and you can find them in the supplement.

With kind regards, Lena Boysen & co-authors

Please also note the supplement to this comment:
https://bg.copernicus.org/preprints/bg-2020-229/bg-2020-229-AC4-supplement.pdf
* * *
[Figure]

**Supplement:**

**Dear Referee,**

**thank you for providing us with this constructive review.**

This study reports on the multi-model LUMIP deforestation experiment. The authors show that controlled, large-scale, global deforestation may contribute global geophysical cooling of near-surface temperatures and global geochemical warming. The geo- physical effects vary latitudinally and by model, generally with warming in the tropics and cooling elsewhere, while the geochemical effects are estimated offline as warming everywhere. The geochemical effects generally are greater than the geophysical effects, leading to net warming, although a potential CO2-enhanced land sink is not included here. Land carbon losses are driven by vegetation loss. Some novel metrics for assessing and potentially estimating the effects of deforestation are also presented.

I appreciate the tremendous effort the authors have put into this study to advance our understanding of the effects of land cover change on the earth system. I have a few main comments, followed by some brief detailed comments.

**Thank you for your acknowledgement of our work and your valuable feedback to our manuscript. In the following I will address your points and how we want to improve the manuscript.**

1. Improving the abstract:

What are the take home messages? There are a lot of different analyses, and only one aspect is highlighted in the abstract. The abstract includes some key numbers, and at least the potential net warming statement of large scale deforestation. While this in itself is a key finding (with the associated caveat of constant CO2), there are a few other notable results to highlight. Missing are the policy/scenario implications related to time/fraction of emergence for climate vs carbon. And the potential for rough estimates of response via the sensitivity metrics (which are analogous to climate sensitivity of models). I understand that there are some limitations to the sensitivity metrics and the time of emergence estimates, but based on S13 and S18 (plus the rest of the carbon figures), it seems safe to say that the climate signals have relatively long time frames while carbon signals have relatively short time frames. However, the climate signal emergence is further complicated by observations that show large, immediate meteorological distinctions between forest and grassland. The sensitivity metrics sup- port the temperature and carbon results, and are potentially useful to the community.

**Thank you for pointing this obvious finding out! We will follow your suggestions and make the abstract stronger and more significant by including these results.**

2. Swap figures

In relation to comment (1) above, switching some of the regular and supplemental figures would make the paper stronger. For example, figures S22 (carbon sensitivity to deforestation) and S18 (ToE and FoE) are more relevant to the carbon points than figures 9-10, which are explanatory. Also, figure S3 is much clearer and easier to understand than figure 3, and follows the text better (you can add Tas to fig S3).

**We agree that some of the figures have the potential to be shown in the main text. However, we would like to keep the number of figures as it is now and the decision for one or the other figure is difficult. We would like to keep the GPP and carbon time series figures in the main text as they show fundamental results. The decision for the surface energy balance decomposition figure has also more sides to it. We argue that the model-wise comparison in the manuscript offers an easier access to the model's performance to**

simulate T$_{surf}$**. On the other hand, a component-wise presentation provides a better inter-model comparison which might be more valuable to the reader. We therefore agree to swap Figures 3 and S3.**

3. Temperature definitions
The descriptions of Tsurf-model and Tas are not complete, which makes it difficult to properly assess the temperature comparisons. While Tsurf is clearly a radiative temperature, is Tsurf-model a radiative temperature for all the models? Tsurf in some models is a canopy air temperature, at a height dictated by the displacement height and aerodynamic roughness. The 2m air temperature is often the air temperature 2m above this "Tsurf." It is important to be clear as to what and where these temperatures actually are, as shown by some of your references.

**Thank you for this inquiry. We will add more specifications to the method and results section on where surface temperature is calculated in the models and what this implies.**

4. MIROC
MIROC does not seem to meet the deforestation harmonization requirements, and its plots don't seem to add to the understanding of the issue. In fact, a lot of extra text is dedicated to explaining why MIROC is different from the others. It would be cleaner if it were not included.

**We would like to keep MIROC in the analysis. We list the caveats of each model's execution of the experiment and highlight if this is the reason for an observed result. This in itself is a demonstration of the difficulties in carrying out harmonized land-use and land cover change-related studies with many participating models. We think that it is still interesting how strong regrowth in this model is and how small the biogeophysical responses are despite a clear disturbance. Furthermore, MIROC simulates interesting carbon dynamics and should therefore be accounted for. We argue that it would be more inclusive to leave MIROC in all analyses than to keep it in only some relevant ones (e.g. the carbon analysis).**

**We will add 'We nevertheless analyse results from MIROC to not only demonstrate the effect these different technical realizations of one scenario can have but to also to draw conclusions for improvements in this model.'**

5) Some supplemental figures are cited out of order.
**We will fix that. Thank you for your close observation.**

**Thank you for your further comments to improve the readability and understanding of our study. We will take them into account during the revision (see the detailed list below).**

**With kind regards,**
**Lena Boysen & co-authors**

Lines 229-227: With respect to the effects of differences in initial forest cover on the implementation of deforestation (lines 229-227), the author's may be interested in this recently published paper: A.V. Di Vittorio, X. Shi, B. Bond-Lamberty, K. Calvin, A. Jones, 2020, "Initial land use/cover distribution substantially affects global carbon and local temperature pro- jections in the integrated Earth system model", Global Biogeochemical Cycles. doi: 10.1029/2019GB006383.
**Reply: We have already cited this study in line 236.**

Lines 326-327: his isn't clear from fig 3. fig s4 is more appropriate here.
**Reply: Yes, we will add 'and Fig. S4c'.**

Line 465: which figures show these regional effects?
**Reply: We will add '(not shown)'.**

Line 505: relate toe and implications to observation of immediate temperature differences between forest and grassland, and perceived differences
**Reply: Immediate effects of deforestation are difficult to capture as variability makes it hard to pin them down. Even the time series of temperature have a 30-year moving average applied and only start after 15 years. We are therefore afraid to not be able to meet this request.**

Line 536: this paragraph is out of place - it doesn't relate to the rest of the section
**Reply:  We will shift the paragraph to the beginning of this section.**

Lines 644-646: fig S19 shows declines in GPP for CESM throughout the deforestation area, so it isn't clear how CESM has increases in GPP where the other model have decreases.
**Reply:  Fig S19 shows the ToE for GPP. We will add, that changes in GPP are seen in 'Fig. 9'. Relating these to Fig. S19 makes sense.**

Lines 680-684: based on fig S20, it doesn't appear the MIROC can have the highest sensitivity. most of its coverage has the smallest change in c per fraction of deforestation.
**Reply: Only regionally, MIROC reaches high carbon losses per deforestation fraction. Globally, MIROC is at the low end across models. We will add '…in boreal North America.'**

Lines 710-713: this is unclear - you have separated your total range in two, arbitrarily, and included veg only change models in one group with other total land c change models.
**Reply: We will delete 'The remaining models would yield a warming of between 0.24 to 0.87 °C.' and changed the previous sentence to 'For MIROC, IPSL and MPI (0.18 to 0.57 °C) this is the main temperature response to ΔF (with only non-significant BGP-induced effects)…'**

---

## Author Comment (AC5) · 7 Sep 2020

Dear David Lapola,

Attached you can find additional figures regarding surface temperatures and heat fluxes as mentioned in the author comments. The responses are minor except for the Arctic regions and the 'warming blob'.

With kind regards, Lena Boysen
* * *
[Figure]

**Fig. 1.** Surface temperatures

[Figure]

**Fig. 2.** Latent heat fluxes

[Figure]

**Fig. 3.** Sensible heat fluxes

---

## Author Response (AR1)

Max-Planck-Institut für Meteorologie | Bundesstr. 53 | 20146 Hamburg

**Dr. Lena R. Boysen**
Land im Erdsystem
Max-Planck-Institut für Meteorologie
Bundesstr. 53
20146 Hamburg
Deutschland
Tel.: +49 - (0)40 - 41173 - 293
lena.boysen@mpimet.mpg.de
www.mpimet.mpg.de

To

Biogeosciences

Editorial Office

Hamburg, den 21. September 2020

Dear Alexey V. Eliseev,

we carefully revised the manuscript following the comments and recommendations of the three referees. You can find a list of the main and minor points below. We implemented all the indicated edits and corrections as stated in the author responses and include a manuscript version with tracked changes hereafter.

EC-Earth's simulation data will hopefully be uploaded to ESGF until the end of this week. We would therefore kindly ask for the permission to add the DOI of the EC-Earth simulations to the SI and to change one sentence about the data availability of this model in the main text in a further version of this manuscript within the next couple of days.

With kind regards,

Lena Boysen and co-authors

**Main comments**

**Referee #1: How do results change if only those models were included that followed the protocol closely?**

We added to line 677: If only models were included that followed the protocol closely (MPI, CNRM and CESM2), the multi-model mean would be -282 GtC after 50 years and -300 GtC after 75 years.

**Referee #1: Is it worth to keep MIROC in the analysis?**

We added to line 262: We nevertheless analyse results from MIROC to not only demonstrate the effect these different technical realizations of one scenario can have but to also to draw conclusions for improvements in this model.

**Referee #2: What is the influence of the ocean?**

We have added more information on the 'warming blob' observed in the Atlantic Ocean (line 375), explained the importance of the ocean in coupled ocean-atmosphere simulations with land-use more (line 83) and highlighted the influence of the ocean with regard to the detection of the climate signal (line 588).

**Referee #3: Improving the abstract**

We added the policy relevance of the long detection time and the time of emergence of carbon changes to the abstract. Further, we now introduce the forest sensitivity in the abstract to highlight this analysis more.

**Referee #3: Swapping figures of the main text and supplementary.**

We exchanged Fig. 3 with Fig. S3 which indeed follows the story line better. We added multi-model means and removed MIROC from the surface energy balance decomposition analysis. We kept all the other figures as they were as we find them better suited for the main text.

**Referee #3: Incomplete definitions of temperatures.**

We now list how different models simulate surface temperature or calculate near-surface temperature at 2m (lines 210 and 220-224). However, we also highlight that these different definitions do not fully explain the gap between the calculation and simulates surface temperatures in Fig. S3 (line 437).

**Referee #3: Exclusion of MIROC.**

As described before we liked to keep MIROC in the analysis. However, we removed it from the surface energy balance decomposition analysis (Fig. 3) as hardly any changes could be observed.

**Referee #1:**

Line 107: supplementary figure 2 is cited before figure S1. Invert figures S1 and S2.
**Reply**: We changed the order of the figures in the SI.

Line 127: "kilometres" should be "kilometers".
**Reply**:  We corrected this.

Lines 150-161: Six out of nine models present some divergence from the protocol. If the analysis is performed only on the remaining three models, is it changing the main results?
**Reply**: We added the results for only the three models that correctly followed the protocol.

Line 163: How many members does MPI provide? Seven or eight?
**Reply**: MPI provided seven members; we corrected this in line 206.

Line 170: the "branching year" is not given for all models in Table S1.
**Reply:** We added the branching year for all models.

Equation 1: Tsurf,piControl should be $T_{surf,piControl}$ with "surf,piControl" subscript.
**Reply:** We corrected the subscript.
Table 1: which symbol denotes non-significant changes (it is missing in my pdf version)?
**Reply:** As stated in the caption, non-significant values are shown in parentheses.

Table 1: does "Pr" stand for precipitation?
**Reply:**  We added that 'Pr' stands indeed for precipitation.

Table 1: in "DcLand over DF" of "MPI" there is an extra parenthesis ")".
**Reply:** We deleted the parenthesis around MPI.

Figure 2: "near surface air temperature" refers to DTas?
**Reply:** We added '(ΔTas)' to the caption.

Figure 3: why don't you directly use Figure S3 as Figure 3? Figure S3, indeed, contains the same information in Figure 3, but it also shows the global versus deforested grid-cell comparison. In figure S3, you could add the multi-model mean values, as done in figure 5.
**Reply:** We swapped Fig. 3 and S3 as another referee also requested. We see the value in both figures which add different messages to the manuscript.

Lines 323-324: where can I see the changes in evapotranspiration and latent heat flux?
**Reply:** ET is shown in Fig. S8, latent heat fluxes are not displayed in a map. We added the link and '(not shown)' to the text.

Line 326: "net shortwave", but the brown line in Figure 2 shows incoming shortwave and not the net one.
**Reply**: We added that this can better be seen in Fig. S4c.

Line 327: the temperate and boreal areas are introduced, but the latitudinal range of these regions are only reported later on in the text (caption of Figure 4).
**Reply:** We added the latitudinal range.

Line 328: Figure S5 shows the distribution of albedo changes. It is not showing that "increasing albedo are stronger than reduction in longwave radiation".
**Reply:** We shifted 'Fig S5' to right behind the mentioning of 'albedo'.

Lines 328-329: "net shortwave reduces", is it shown in figure S4c?
**Reply:** We added '(Fig. S4c)'.

Line 329: "incoming shortwave radiation increases", is it brown line in Figure 3?
**Reply:** Yes, it is. We added this.

Lines 330-331: "MPI and IPSL reduction in cloud cover", is it shown in Figure S4f?
**Reply:** Yes, we added the link.

Line 332: "turbulent heat flux", is it shown in Figure S4a?
**Reply:** Yes, we added the link.

Line 342: "increase in albedo", is it visible in Figure S5?
**Reply:** This should be 'increase in the albedo… (Fig. 5)'.

Line 343: "reduced cloud cover", is it shown in Figure S6?
**Reply:** Yes, we added the link.

Line 343-344: where is shown the comparison between changes in incoming shortwave from clouds and incoming shortwave from albedo?
**Reply:** This was inferred by comparing the contributions of albedo and incoming shortwave radiation itself on Tsurf. For IPSL, CNRM, Can and BCC, the incoming shortwave radiation contributes with a greater increase in Tsurf than the reduction caused by albedo does. We added the references to the text.

Line 347: "as also observed" should be "is also observed".
**Reply**: This is  corrected.

Line 352: "net warming", is it visible in Figure 2c?
**Reply:**  Yes, and Fig. 3c. We added the link.

Line 352: "evaporative cooling", is it shown in Figure S8i?
**Reply:** Yes, and Fig. S3i. We added the links.

Line 354: "sensible heat fluxes", is it shown as cyan lines in Figure 3?
**Reply:** Especially in Fig S3d and S4e. We added the link.

Line 355: "BCC temperature increase", is it visible in Figure 2f?
**Reply:** Yes, and S3f. We added the link.

Line 359: "evaporation decrease", is it shown in Figure S8f?
**Reply:** Yes! We added the link.

Line 359: "increase cloud formation", the cloud increase seems limited over the Amazon in Figure S6f.
**Reply:** Yes, but the text passage only deals with this region, so this should be covered.

Line 361: "not shown", the difference between longwave and longwave clear sky is displayed in Figure S7. Are you referring to different variables? Otherwise, the difference is shown. The described difference is not visible to me in Figure S7.
**Reply:** Indeed, this is shown in Fig. S7 but hard to detect. This is a very local region in northern Amazon but which leads to a strong temperature signal.

Line 364: is CNRM case shown in Figure 3e?
**Reply:** Yes, we added the link.

Lines 369-370: formatting error: the new line is started too early.
**Reply**: This is corrected.

Line 383: available energy should be Figure S4b instead of Figure S4a.
**Reply:** Yes, indeed.

Line 385: "cloud formation", is it visible in Figure S4f?
**Reply:** Yes, we added the link.

Line 386: turbulent heat flux should be Figure S4a instead of Figure S4b.
**Reply:** Yes, indeed.

Line 386: is latent heat flux visible in Figure S4d?
**Reply**: Yes, and S8.

Line 388: "[...] over temperate and boreal over grassland [...]", the second over should be deleted.
**Reply:** This is corrected.

Lines 416-421:       The temperature changes per unit fraction of grid cell deforested range between 4 and -20, why Figure S9 has values between 2 and -2?
Similarly, the temperature changes per unit area deforested reported in Figure S10 and the text cover a different range of values.
**Reply:** Only a few pixels show these extreme values; If we extended the color bar range we would lose the visibility of the main range. We added '(Note that the color bar range is limited and extreme values not shown).'

Line 431: "On average [...] 0.27 °C frac$^{-1}$", does it refer to Figure 4a?
**Reply:** Yes, we added '(derived from Fig. 4a)'.

Line 434: "reverse at higher DF", does it refer to Figure 4c?
**Reply:** Yes, we added the link.

Line 434: "regions_there" should be "regions (Fig. 4b) there".
**Reply**: This is corrected and the link added.

Lines 465-467: "Over South America [...] over Eurasia", where are these changes shown?
**Reply:** The regional time series is not shown. We have added 'not shown' to the text.

Line 485: "30% and 10% of deforestation still left", is it visible in Figure S12?
**Reply:** Yes, we added the link.

Line 506: "2013) and" should be "2013; and".
**Reply:** This is corrected.

Lines 572-573: "For all models but MIROC [...] (DcSoil).", is it visible in Figure 8?
**Reply:** Yes, we added the link.

Lines 577-579: "Below ground carbon [...] several decades)", are fast and medium soil carbon pools displayed in Figure S17?
**Reply:** Yes, we added the link.

Line 585: "large NEP", is it visible in Figure 10d?
**Reply:** Yes, we added the link.

Lines 594-595: are heterotrophic respiration and net ecosystem productivity shown in Figure 10?
**Reply**: Yes, we added the links.

Line 598: "NPP reduction", is it depicted in Figure 10b?
**Reply:** Yes, we added the link.

Lines 617-618: "cVeg change [...] carbon pools", is it visible in Figure 8?
**Reply:** Regional soil carbon time series are not shown. We added this to the text.

Lines 618-619: "The exception [...] soil carbon pool", where is it shown?
**Reply:** We added '(not shown)'. Listing all regional time series would be too much for the SI.

Lines 621-622: "In BCC, [...] simulation period", is it shown in Figure 10?
**Reply:** Yes, we added the link.

Line 628: "uniform global increase in NPP", is it visible in Figure 10b?
**Reply:** Yes, we added the link.

Lines 628-629: "especially in the tropics", where is it shown?
**Reply:** In Fig. 9d. We added the link.

Lines 630-631: "GPP changes are [...] south-east Asia", is it shown in Figure 9b?
**Reply:** In Fig. 9d. We added the link.

Lines 634-636: NBP values are the ones shown in Figure 10f?
**Reply**: Yes, we added the link.

Lines 644-645: "GPP reductions [...] GPP increase in time", are the changes shown in Figure 9?
**Reply:** Yes, we added the link.

Lines 680-684: The range of values displayed in Figures S20 and S21 are different from the values reported in the text. For example, Figure S20 shows values between -20 and 20 kg m-2 frac-1, while the text describes changes between -40 and -90 GtC.
**Reply:** The range displayed in the color bar is not the complete range. Adding the extreme values would lead to less visibility of the main range.

Lines 686-687: "Fig. S25" should be "Fig. S22".
 **Reply**: This is corrected.

Lines 691-692: "In the tropics, IPSL [...] forest removals.", is it visible in Figure S22a?
**Reply:** Yes, we added the link.

Line 695: "removal in South America", is it shown in Figure S20c?
**Reply**: It's S23c. We added the link.

**Conclusions**: the conclusions should also report the opposite biogeophysical response between boreal/temperate forests and tropical forests. The possibility to identify a "threshold" latitude below which re/afforestation could have both biogeophysical and biogeochemical cooling effect is significant.
**Reply**: We added 'On average, the switch of sign from tropical warming to extra-tropical cooling happens around 22.6°N.'.

Figure S2: caption: "Mkm2" should be "Mkm$^2$" with "2" as apex.
**Reply**: This was an issue of converting the .docx to PDF. It is fixed.

Figure S7: why doesn't this figure report CESM2?
**Reply:** CESM2 did not provide clear-sky longwave radiation. We have added this to the caption.

Figure S9: caption: "frac-1" should be "frac$^{-1}$" with "-1" as apex.
**Reply**: This was an issue of converting the .docx to PDF. It is fixed now.

Figure S10: caption: "km2" should be "km$^2$" with "2" as apex.
**Reply:** This was an issue of converting the .docx to PDF. It is fixed.

Figure S13: "Only statistically significant areas as found in Fig. 2 are shown." but the figure's legend report also "non-sign.". Are non-significant areas plotted or not?
**Reply**: True, we only display significant values. We corrected the legend.

Figure S15: caption: "m2" should be "m$^2$" with "2" as apex. Add why some models are not plotted.
**Reply:** This was an issue of converting the .docx to PDF. It is fixed.

Figure S16: caption: "km2 year-1" should be "km$^2$ year$^{-1}$" with "2" and "-1" as apex. Add why some models are not plotted.
**Reply:** This was an issue of converting the .docx to PDF. It is fixed.

Figure S18: caption: "ToE in years [...]" should be "ToE and FoE in years [...]".

**Reply:** It should actually be 'ToE in years and the corresponding FoE in %...'.

Figure S19: caption: "Same as Fig S19 but for the ToE of gross primary productivity" should be "Same as Fig S18 but for the ToE and FoE of gross primary productivity".
**Reply:** This is corrected.

Figure 20-23: caption: Add why some models are not plotted.
**Reply:** We added to Fig. S15, S20, S21 and S23: 'Note, that CanESM and BCC did not follow the protocol when removing forests (see section 2.2) and are therefore not displayed for land carbon pools.'
* * *
**Referee #2:**

Line 51: future losses in what time horizon?
**Reply: We added 'until the end of the century'.**

Line 52: not only to free land for food and bioenergy but to speculate on land ownership itself (i.e. to justify land grabbing [see Rajão et al. 2020 Science]).
**Reply: Yes, that is true. However, to our knowledge this is not included in Hurtt et al. (2020) and therefore we would like to leave it as it is.**

Line 70: "An experimental setup with atmosphere-only models in which sea surface tem- peratures are prescribed allows to increase the signal-to-noise ratio of models' re- sponse to deforestation. This setup assumes that the effect of large-scale circulation changes is small and can be ignored." Please explain this better. You mean the sen- sitivity of models to deforestation increases (artificially?) in atmosphere-only model setups? "(. . .) the effect of large-scale (atmospheric?) circulation changes is small (compared to what?) and can be ignored."
**Reply: We added: 'The cooling of the of the land surface via enhanced albedo cools and dries the whole troposphere which in turn transfers this signal via reduced longwave radiation further to the ocean. With prescribed SSTs the mediating effect of the ocean on the land temperatures is missing resulting in overestimated tropical warming and underestimated boreal cooling (Davin and de Noblet-Ducoudré, 2010).'**

Line 125: if tree and forest are different things why using them interchangeably then?
**Reply: We changeed all 'tree fractions' or 'tree area' to 'forest fractions' or 'forest area'. We deleted the line 'Note, that we here use the terms tree and forest fraction interchangeable, although they are defined distinctively in reality.'**

Table 1: are you sure the units of delta Pr is mm/yr?
**Reply: Yes.**

Line 323: is this somewhat related to an acceleration of the Atlantic THC? By the way that is the only region the IPCC shows as cooling in the SRES scenarios. Please explore the implications of this result deeper.
**Reply: We added: 'This result is in line with the reversed finding by Rahmstorf et al. (2015) who found a 'cooling blob' due to the freshwater input from the Greenland ice shield caused by global warming slowing down the meridional overturning circulation.'**

Line 353: the increase in leaf area is such that LAI surpasses that of a typical tropical forest?
**Reply: Yes, since the regrowth of forest is suppressed in this setup.**

Line 359: But this also decreases incoming shortwave radiation, doesn't it? Fig 3f doesn't show a considerable increase in longwave radiation.
**Reply: This observed phenomena is very local and averaged out in Fig. 3. You would see it in a map of shortwave radiation which is not shown and you can actually see in in Fig. S6 and S7. We added this to the text.**

Line 378: a "global" cooling effect due to CO2 removal. . .
**Reply: We added this.**

Line 379: uncertain
**Reply: This is corrected.**

Lines 389-390: Is this based on in-field experimental studies or modeling? How trustful is it?
**Reply: Winckler et al. (2019b) performed model simulations with the MPI-ESM but also compared the results with various observational data sets which capture by nature only local effects.**

Line 414: Tas or Tsurf?
**Reply: 'Tas'**

Fig. 6a: ToE in the tropics seems restricted to 30 years or more (i.e. values smaller than 30yr do not occur). Is that due to the use of a 30-yr moving average?
**Reply: If you zoom into IPSL or CESM2 you can recognize single pixels with brighter colors / years below 30. The data was processed with the same script (no moving average applied!) so we can conclude that no robust effects appear in MPI before 30 years.**

Line 501: ". . .robustly detectable after a few decades". What is that time lag attributable to? Influence of slower thermal sensitivity of the adjacent oceans? That's one strong reason to show sea surface temperature and circulation changes in this article.

**Reply: We added '… decades as climate variability and mediating effects from the ocean have to be overcome (Davin and de Noblet-Ducoudré, 2010).'**

Lines 508-509: but notice that this study did not test that (i.e. the climatic effects of re/afforestation). Would be a nice follow-up.
**Reply: We changed 'are' to 'could be'.**

Line 523: What is an unfavourable climatic condition? I suggest a different term also on L546.
**Reply: We added '(too dry or hot)'.**

Fig. 9: after how many years?
**Reply: Like all maps as described in the methods section 2.3 first sentence: Climatic variables cover the last 30 years, carbon variables the last 10 years. We added 'averaged over year 50 to year 79' or 'averaged over year 70 to year 79' to every spatial plot.**

Line 591: I think the authors should present arguments why the C4 grasses are "overly" productive. C4 grasses like sugarcane or others can indeed have a NPP higher than that of tropical forests.
**Reply: C4 grasses are parametrized in a way that they survive in arid regions. When entering the tropics, they are therefore too productive. This is laid out in line 523 (see above).**

Line 621: It is unclear why figure 9f shows a steep decline of GPP in northern Amazon, considering that considerable increase in precipitation for that specific region was pre- viously presented in the paper for BCC.
**Reply: We added 'Only in northern Amazon region, GPP is reduced under high temperatures and despite the observed precipitation increase (Fig. 7).'**

Line 765: Why wasn't this analysis of large-scale circulation patterns done in this study?
**Reply: Such a thorough study deserves its own publication and would exceed the scope for this manuscript (BGP and BGC effects).**

Line 767: ". . .temperature gradient mau alter the large-scale circulation. . ." Atmospheric or oceanic?
**Reply: We added 'atmospheric'.**

Line 774: Is it possible to be a bit more specific on the sort of observational data needed?

**Reply: We have listed Duveiller et al. (2018b) who created a spatially and temporally explicit data set based on various observational data sets (ESA CCI and MODIS) to investigate the non-local effects to land cover change.**

Fig. S13: the brownish color in the tropics in panel f and in the boreal zone in panel d doesn't seem to correspond to any color in the legend.
**Reply:  This is fixed.**
* * *
**Referee #3:**

Lines 229-227: With respect to the effects of differences in initial forest cover on the implementation of deforestation (lines 229-227), the author's may be interested in this recently published paper: A.V. Di Vittorio, X. Shi, B. Bond-Lamberty, K. Calvin, A. Jones, 2020, "Initial land use/cover distribution substantially affects global carbon and local temperature pro- jections in the integrated Earth system model", Global Biogeochemical Cycles. doi: 10.1029/2019GB006383.
**Reply: We have already cited this study in line 236.**

Lines 326-327: his isn't clear from fig 3. fig s4 is more appropriate here.
**Reply: Yes, we added 'and Fig. S4c'.**

Line 465: which figures show these regional effects?
**Reply: We added '(not shown)'.**

Line 505: relate toe and implications to observation of immediate temperature differences between forest and grassland, and perceived differences
**Reply: Immediate effects of deforestation are difficult to capture as variability makes it hard to pin them down. Even the time series of temperature have a 30-year moving average applied and only start after 15 years. We are therefore afraid to not be able to meet this request.**

Line 536: this paragraph is out of place - it doesn't relate to the rest of the section
**Reply:  We shifted the paragraph to the beginning of this section.**

Lines 644-646: fig S19 shows declines in GPP for CESM throughout the deforestation area, so it isn't clear how CESM has increases in GPP where the other model have decreases.
**Reply:  Fig S19 shows the ToE for GPP. We added, that changes in GPP are seen in 'Fig. 9'. Relating these to Fig. S19 makes sense.**

Lines 680-684: based on fig S20, it doesn't appear the MIROC can have the highest sensitivity. most of its coverage has the smallest change in c per fraction of deforestation.

**Reply: Only regionally, MIROC reaches high carbon losses per deforestation fraction. Globally, MIROC is at the low end across models. We added '…in boreal North America.'**

Lines 710-713: this is unclear - you have separated your total range in two, arbitrarily, and included veg only change models in one group with other total land c change models.

**Reply: We deleted 'The remaining models would yield a warming of between 0.24 to 0.87 °C.' and changed the previous sentence to '
[revised manuscript text omitted]